# A potent voltage-gated calcium channel inhibitor engineered from a nanobody targeted to auxiliary Ca$_V$β subunits

Travis J Morgenstern[1], Jinseo Park[1], Qing R Fan[1], Henry M Colecraft[1,2]*

[1]Department of Pharmacology, Columbia University, Vagelos College of Physicians and Surgeons, New York, United States; [2]Department of Physiology and Cellular Biophysics, Columbia University, Vagelos College of Physicians and Surgeons, New York, United States

**Abstract** Inhibiting high-voltage-activated calcium channels (HVACCs; Ca$_V$1/Ca$_V$2) is therapeutic for myriad cardiovascular and neurological diseases. For particular applications, genetically-encoded HVACC blockers may enable channel inhibition with greater tissue-specificity and versatility than is achievable with small molecules. Here, we engineered a genetically-encoded HVACC inhibitor by first isolating an immunized llama nanobody (nb.F3) that binds auxiliary HVACC Ca$_V$β subunits. Nb.F3 by itself is functionally inert, providing a convenient vehicle to target active moieties to Ca$_V$β-associated channels. Nb.F3 fused to the catalytic HECT domain of Nedd4L (Ca$_V$-aβlator), an E3 ubiquitin ligase, ablated currents from diverse HVACCs reconstituted in HEK293 cells, and from endogenous Ca$_V$1/Ca$_V$2 channels in mammalian cardiomyocytes, dorsal root ganglion neurons, and pancreatic β cells. In cardiomyocytes, Ca$_V$-aβlator redistributed Ca$_V$1.2 channels from dyads to Rab-7-positive late endosomes. This work introduces Ca$_V$-aβlator as a potent genetically-encoded HVACC inhibitor, and describes a general approach that can be broadly adapted to generate versatile modulators for macro-molecular membrane protein complexes.

DOI: https://doi.org/10.7554/eLife.49253.001

*For correspondence:
hc2405@cumc.columbia.edu

Competing interests: The authors declare that no competing interests exist.

## Introduction

Inhibition of high-voltage-activated calcium channels (HVACCs) is an important prevailing or potential therapy for diverse cardiovascular (hypertension, cardiac arrhythmias, cerebral vasospasm) and neurological diseases (epilepsy, chronic pain, Parkinson's disease) (*Zamponi et al., 2015*). Small molecule HVACC inhibitors include Ca$_V$1 blockers (dihydropyridines, benzothiazepenes phenylalkylamines) and venom peptides that target Ca$_V$2.1 ($\omega$-agatoxin), Ca$_V$2.2 ($\omega$-conotoxin), and Ca$_V$2.3 (SNX-482) channels. When introduced into an organism, small-molecule HVACC blockers are typically widely distributed leading to off-target effects that can narrow the therapeutic window and, thereby, adversely impact therapy. Genetically-encoded HVACC inhibitors can circumvent off-target effects because they can be selectively expressed in target tissues or cells; thus, they may be useful alternatives or complements to small molecule therapy (*Yang et al., 2013*; *Murata et al., 2004*).

There are seven distinct HVACCs (Ca$_V$1.1 - Ca$_V$1.4; Ca$_V$2.1 - Ca$_V$2.3) which exist in cells as multi-subunit complexes comprising pore-forming $\alpha_1$-subunits assembled with auxiliary proteins which include β, $\alpha_2$-δ, and γ subunits (*Zamponi et al., 2015*; *Buraei and Yang, 2010*; *Dolphin, 2012*). HVACCs are named according to the identity of the component $\alpha_1$ subunit ($\alpha_{1A}$-$\alpha_{1F}$; $\alpha_{1S}$) which also contains the voltage sensor, selectivity filter, and channel pore. The various auxiliary subunits typically regulate HVACC trafficking, gating, and modulation, and are recognized as potential targets for developing HVACC-directed therapeutics. For example, gabapentin, which is clinically utilized

for treating epilepsy and neuropathic pain, targets HVACC $\alpha_2$-$\delta$ subunits (*Gee et al., 1996*). Based on the presumption that the association of $\alpha_1$ with $\beta$ is obligatory for the formation of surface-targeted functional HVACCs as indicated by heterologous expression experiments (*Buraei and Yang, 2010*), disruption of the $\alpha_1$-$\beta$ interaction has been long pursued as a strategy to develop HVACC inhibitors (*Young et al., 1998*; *Findeisen et al., 2017*; *Chen, 2018*; *Khanna et al., 2019*). To this end, over-expression of peptides derived from the $\alpha_1$-interaction domain (AID) which contains the amino acid sequence responsible for high-affinity $\alpha_1$-$\beta$ association (*Pragnell et al., 1994*; *Van Petegem et al., 2004*; *Chen et al., 2004*; *Opatowsky et al., 2003*), has been utilized by several groups as putative genetically-encoded HVACC inhibitors (*Findeisen et al., 2017*; *Yang et al., 2019*). However, the efficacy of this approach in vivo may be limited as recent data suggests that in some adult tissue the $\alpha_1$-$\beta$ interaction is not absolutely essential for surface trafficking of HVACCs (*Yang et al., 2019*; *Meissner et al., 2011*).

Rad/Rem/Rem2/Gem/Kir (RGK) proteins are endogenous small Ras-like G-proteins that profoundly inhibit all HVACCs when over-expressed in either heterologous cells or native tissue (*Béguin et al., 2001*; *Finlin et al., 2003*; *Chen et al., 2005*; *Xu et al., 2010*). They form ternary complexes with HVACCs via binding to constituent $\beta$ subunits and inhibit currents via multiple mechanisms including removal of surface channels and impairing gating (*Yang and Colecraft, 2013*; *Yang et al., 2010*). Despite their efficacy, utility of RGKs as genetically-encoded HVACC inhibitors is confounded by potential off-target effects since they interact with and regulate other binding partners such as cytoskeletal proteins, 14-3-3, calmodulin, and CaM kinase II (*Yang and Colecraft, 2013*; *Correll et al., 2008*; *Royer et al., 2018*; *Béguin et al., 2005*; *Ward et al., 2004*). A critical unmet need is the development of genetically-encoded HVACC inhibitors that possess the high efficacy of RGKs but lack the problematic interactions with other signaling proteins. Here, we achieve this by fusing the homologous to the E6-AP carboxyl terminus (HECT) catalytic domain of the E3 ubiquitin ligase, neural precursor cell developmentally down-regulated protein 4 (Nedd4-2 or hereafter referred to as Nedd4L), to a Ca$_V$$\beta$-targeted nanobody. The resulting construct, termed Ca$_V$-a$\beta$lator, eliminated diverse HVACCs both in both reconstituted systems and native excitable cells, providing a unique new tool for probing Ca$_V$1/Ca$_V$2 signaling and regulation in vivo, and potential development into a therapeutic.

## Results

### Isolation and characterization of Ca$_V$$\beta$-targeted nanobodies

We sought to develop a nanobody targeted to Ca$_V$$\beta$s that would be incorporated into Ca$_V$ channel complexes but be functionally silent, to serve as a vehicle to potentially address distinct enzymatic moieties or sensors to endogenous channels. We expressed Ca$_V$$\beta_{1b}$ and Ca$_V$$\beta_3$ in HEK293 cells using BacMam expression and purified the proteins using affinity purification, ion exchange, and size exclusion chromatography (*Figure 1a*). Purified $\beta_1$ and $\beta_3$ (1 mg each) were used for llama immunization, and successful serum conversion was confirmed by ELISA (not shown). Messenger RNA was extracted from isolated lymphocytes, PCR-amplified and cloned into a plasmid vector (pComb3XSS) to generate a V$_{HHS}$ phage library (*Figure 1b*). Putative nanobody binders were enriched from the phage library using three rounds of phage display and panning (*Pardon et al., 2014*). We performed a 96-well ELISA on enriched phage libraries and selected 14 positive clones for sequencing (*Figure 1c*). We identified at least seven distinct classes of nanobody binders based on the unique sequences within complementarity determining regions (CDR1-3), the major determinants of antigen binding (*Figure 1d,e*).

We adopted a small-molecule-induced fluorescence co-translocation assay to simultaneously determine whether: (1) individual nanobodies were well-behaved when expressed in mammalian cells (i.e. do not aggregate), and (2) bound Ca$_V$$\beta$s. A tripartite construct consisting of individual nanobodies fused to CFP and the C1 domain of PKC$\gamma$ was cloned into a CMV expression vector and transiently co-transfected with YFP-tagged Ca$_V$$\beta$s into HEK293 cells. After pilot experiments, we chose one nanobody clone, nb.F3, for in-depth characterization and development. Both nb.F3-CFP-C1 and YFP-$\beta_1$ were uniformly expressed in the cytosol of transfected HEK293 cells (*Figure 1f*). Application of 1 μM phorbol-12,13-dibutyrate (PdBu) led to the rapid and dramatic redistribution of nb.F3-CFP-C1 from the cytosol to the plasma and nuclear membranes (*Figure 1f*). Reassuringly,

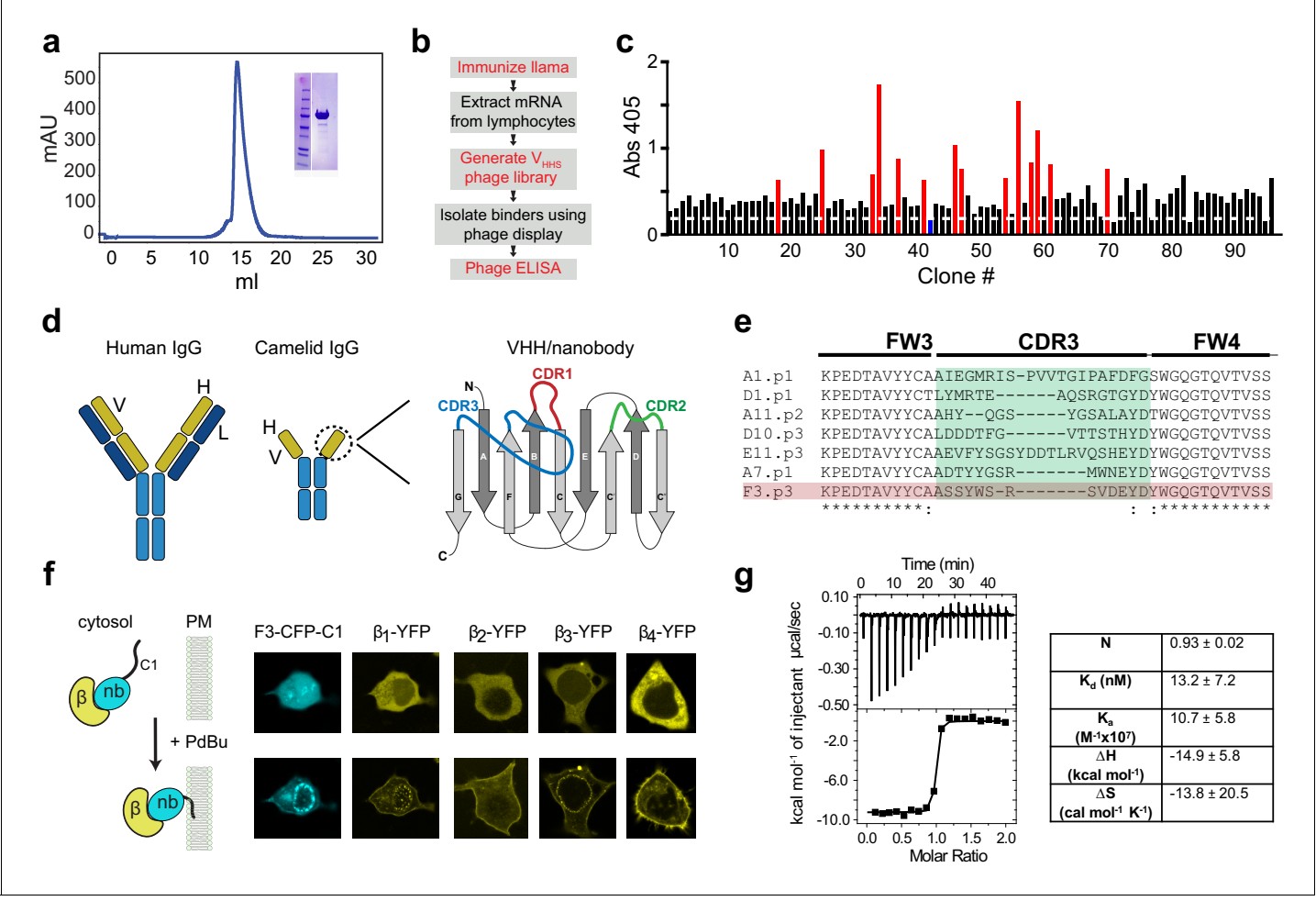

**Figure 1.** Development of a pan-Ca$_V$β nanobody. (a) Size-exclusion chromatograph and Coomassie gel (inset) showing purified Ca$_v$β$_1$ from baculovirus-infected HEK293 GnTl$^-$ cells. (b) Flow-chart of nanobody generation. (c) Phage ELISA using Ca$_V$β$_1$ as bait and periplasmic extracts from single infected *E. coli* clones. Red bars represent clones that were selected for subsequent analyses; blue bar represents a negative control from an *E. coli* expressing an anti-GFP nanobody. (d) Cartoon showing conventional IgG antibody (left) and camelid heavy-chain antibody (center). Right, a schematic representation of the variable heavy chain (VHH or nanobody) of camelid heavy-chain antibodies. The three CDR loops which are the primary determinants of antigen-binding are shown in red, green, and blue. (e) Sequence alignment of CDR3 from selected clones. (f) Left, schematic of co-translocation assay to determine nanobody/Ca$_V$β interaction in HEK293 cells. Right, confocal images showing membrane co-translocation of Ca$_V$β$_X$-YFP and nb.F3-CFP-C1$_{PKCγ}$ in response to treatment with 1 uM phorbol 12,13-dibutyrate (PdBu). (g) Left, exemplar isothermal titration calorimetry trace using purified Ca$_V$β$_{2b}$ and nb.F3. Right, summary of ITC thermodynamic parameters. N, stoichiometry; K$_d$, dissociation constant; K$_a$, affinity constant; ΔH, enthalpic change; ΔS entropic change. T = 298K.

DOI: https://doi.org/10.7554/eLife.49253.002

The following figure supplement is available for figure 1:

**Figure supplement 1.** Nb.F3 binds all four Ca$_V$β subunits in the cytosol of mammalian cells Left, schematic of phorbol ester 12,13-dibutyrate (PdBu) translocation assay.

DOI: https://doi.org/10.7554/eLife.49253.003

YFP-β$_1$ concomitantly redistributed to the plasma and nuclear membranes, providing a convenient visual confirmation that it associates with nb.F3 inside cells (*Figure 1f*). Similar experiments conducted with the other Ca$_V$βs (β$_2$-β$_4$) showed that they all bind with nb.F3-CFP-C1 in cells (*Figure 1f*; *Figure 1—figure supplement 1*), indicating the nanobody interacts with an epitope conserved among Ca$_V$βs. Isothermal titration calorimetry using purified nb.F3 and Ca$_V$β$_{2b}$ indicated a high-affinity (K$_d$ = 13.2 ± 7.2 nM) interaction and a 1:1 stoichiometry (*Figure 1g*).

It was important to our overall strategy that nb.F3 incorporate into assembled HVACC complexes without impacting channel function or subunit stability. We utilized a flow cytometry assay to assess

the impact of nb.F3 on recombinant $Ca_V2.2$ trafficking, subunit expression levels, and whole-cell currents, all of which are known to be regulated by $Ca_V\beta$ (*Figure 2*) (*Waithe et al., 2011*). We used an engineered $\alpha_{1B}$ harboring two tandem high-affinity bungarotoxin-binding sites (2XBBS) in the extracellular domain IV S5-S6 loop and a C-terminus YFP tag to enable simultaneous detection of surface (Alexa647-conjugated bungarotoxin) and total (YFP fluorescence) channel populations in non-permeabilized cells (*Figure 2a*). We co-expressed BBS-$\alpha_{1B}$-YFP and $Ca_V\beta$ either with or without nb.F3-

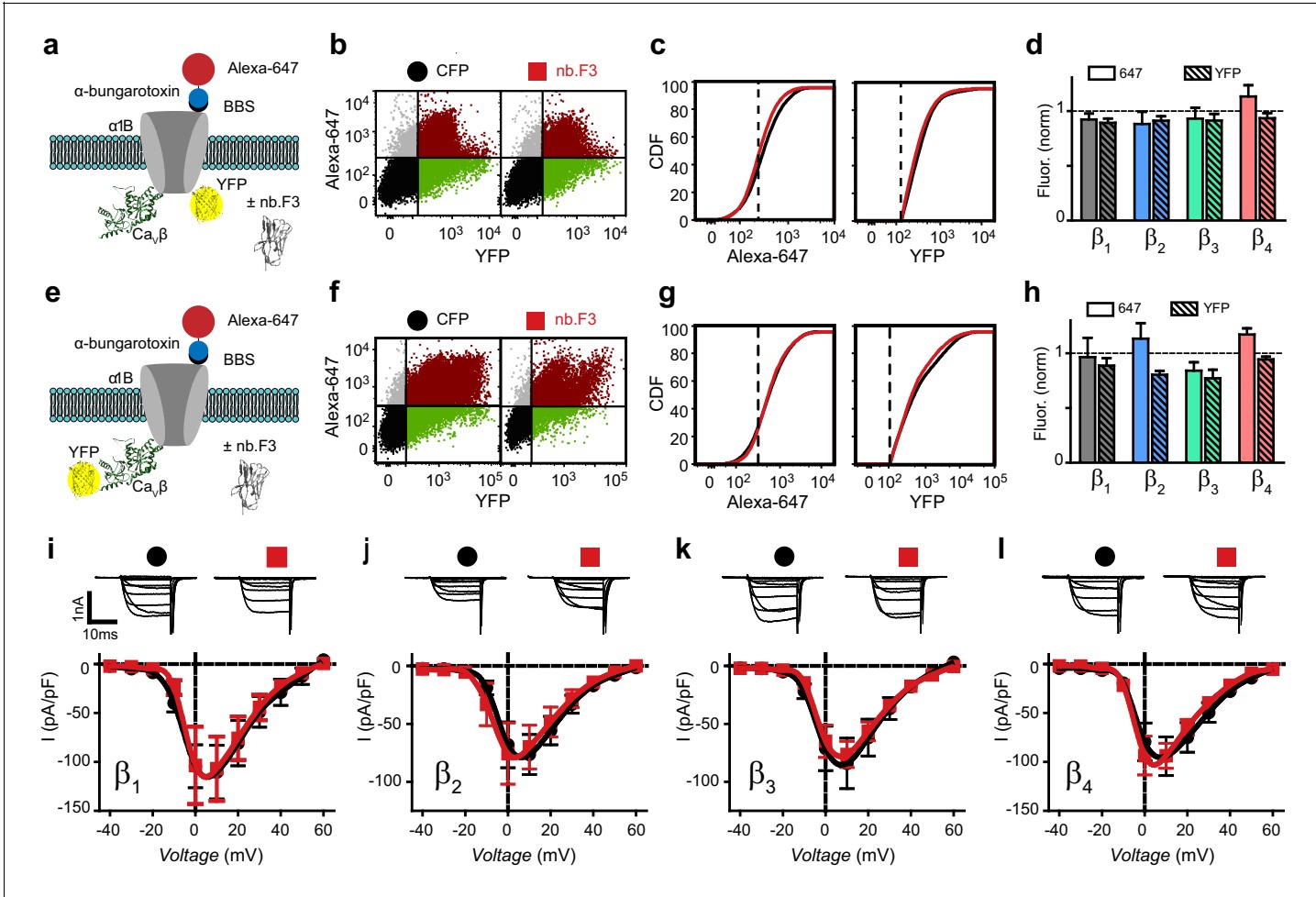

**Figure 2.** Nb.F3 is functionally silent on reconstituted $Ca_V2.2$ channels. (a) Schematic of experimental strategy; BBS-$\alpha_{1B}$-YFP was transfected in HEK293 cells with $\alpha_2\delta$, $Ca_V\beta$ and either CFP or nb.F3-P2A-CFP. (b) Exemplar flow cytometry dot plot of cells expressing BBS-$\alpha_{1B}$-YFP + $Ca_V\beta_1$ + $\alpha_2\delta$-1 and either CFP (left) or nb.F3-P2A-CFP (right). Approximately 100,000 cells are represented here and throughout. Horizontal and vertical lines represent the threshold for YFP- and Alexa-647-positive cells, respectively, as determined with single color controls. (c) Cumulative distribution histogram of Alexa-647 (left) or YFP fluorescence (right) from CFP (black) or nb.F3 (red) expressing cells. YFP-positive cells were selected for the analysis; dashed lines represent thresholds for Alexa-647 and YFP fluorescence signals above background. (d) Summary flow cytometry data of surface (647, filled) and total (YFP, patterned) levels of BBS-$\alpha_{1B}$-YFP. Data from nb.F3-expressing cells was normalized to CFP control group. n=>5,000 cells analyzed per experiment, N=4 separate experiments, error bars, s.e.m. (e) Experimental strategy; HEK293 cells were transfected with BBS-$\alpha_{1B}$ + $Ca_V\beta$-YFP + $\alpha_2\delta$-1. (f-h) Same format as (b-d) for cells expressing BBS-$\alpha_{1B}$ + $Ca_V\beta$-YFP+ $\alpha_2\delta$-1 ± nb.F3-P2A-CFP. (i) Exemplar whole-cell $Ba^{2+}$ currents (top) and population *I-V* curves (bottom) in HEK293 cells expressing $\alpha_{1B}$ + $Ca_V\beta_1$ + $\alpha_2\delta$-1 and either CFP (black) or nb.F3-P2A-CFP (red). (j-l) Same format as (i) for cells expressing $Ca_V\beta_2$ (j), $Ca_V\beta_3$ (k), and $Ca_V\beta_4$ (l). Scale bar 1 nA, 10 ms. Data are means ± s.e.m., n=10 for each point.

DOI: https://doi.org/10.7554/eLife.49253.004

The following figure supplements are available for figure 2:

**Figure supplement 1.** Exemplar flow cytometry data for BBS-$\alpha_{1B}$ with YFP-$Ca_V\beta_2$- $Ca_V\beta_4$.

DOI: https://doi.org/10.7554/eLife.49253.005

**Figure supplement 2.** Nb.F3 is functionally silent on reconstituted $Ca_V1.2$ channels.

DOI: https://doi.org/10.7554/eLife.49253.006

P2A-CFP and utilized flow cytometry to rapidly measure surface and total channel expression. In cells expressing BBS-$\alpha_{1B}$-YFP and $\beta_{1b}$, nb.F3 had no impact on Alexa647 or YFP fluorescence compared to control (*Figure 2b–d*), indicating no disruption of channel trafficking or effect on $\alpha_{1B}$ expression. Similar results regarding the inertness of nb.F3 on $\alpha_{1B}$ trafficking and stability were obtained when Ca$_V$2.2 was reconstituted with the other Ca$_V\beta$ ($\beta_2$-$\beta_4$) subunits (*Figure 2d*).

To examine a potential direct impact of nb.F3 on Ca$_V\beta$ itself, we applied the flow cytometry assay to cells expressing BBS-$\alpha_{1B}$ + $\beta$-YFP ± nb .F3-P2A-CFP (*Figure 2e*). Not surprisingly, nb.F3 did not impact the surface trafficking of BBS-$\alpha_{1B}$ co-expressed with any of the four Ca$_V\beta$ isoforms (*Figure 2f–h*, *Figure 2—figure supplement 1*). The expression levels of $\beta_1$-YFP and $\beta_4$-YFP were unaffected by nb.F3, whereas the levels of $\beta_2$ and $\beta_3$ were modestly reduced (although this effect did not reach statistical significance), suggesting a possible slightly increased vulnerability of these two isoforms to degradation when bound by the nanobody (*Figure 2h*). Similar observations regarding the lack of effect of nb.F3 on channel trafficking and subunit expression levels were made in cells expressing Ca$_V$1.2 channels reconstituted from BBS-$\alpha_{1C}$ + $\beta$-YFP ± nb .F3-P2A-CFP (*Figure 2—figure supplement 2*).

Finally, we used patch-clamp electrophysiology to evaluate the impact of nb.F3 on whole-cell currents through recombinant Ca$_V$2.2 channels reconstituted in HEK293 cells. Cells expressing $\alpha_{1B}$ + $\beta_{1b}$ + $\alpha_2\delta$ displayed robust whole-cell Ba$^{2+}$ currents that were completely unaffected by nb.F3 (*Figure 2i*; $I_{peak,0mV}$ = −104.4 ± 22.0 pA/pF, n = 10 for CFP, and $I_{peak,0mV}$ = −103.5 ± 39.5 pA/pF, n = 10 for nb.F3). A similar lack of effect of nb.F3 was observed on currents from either Ca$_V$2.2 reconstituted with the other $\beta_2$-$\beta_4$ subunits (*Figure 2j-l*), or Ca$_V$1.2 ($\alpha_{1C}$ + $\beta_{2a}$ + $\alpha_2\delta$) channels (*Figure 2—figure supplement 2*).

Overall, these results indicate that nb.F3 binds $\beta_1$-$\beta_4$ subunits in cells, and is potentially assembled into Ca$_V$ channel complexes in a functionally silent manner, essentially acting as an unobtrusive passenger. However, it was also possible that the apparent functional inertness of nb.F3 on Ca$_V$2.2 and Ca$_V$1.2 channels had a more trivial explanation— that Ca$_V\beta$s assembled with pore-forming $\alpha_1$-subunits are simply inaccessible to nb.F3. We could discriminate between these two possible scenarios by determining whether nb.F3 could be used to target bioactive molecules to regulate assembled channels, as we did next.

## Potent functional effects of an F3-Nedd4L chimeric protein on Ca$_V$1/Ca$_V$2 channels

We hypothesized that fusing the catalytic domain of an E3 ubiquitin ligase to nb.F3 would generate a genetically-encoded molecule that inhibits Cav1/Cav2 channels by reducing their surface density (*Kanner et al., 2017*). Accordingly, we generated a chimeric construct (nb.F3-Nedd4L) by fusing the catalytic HECT domain of Nedd4L to the C-terminus of nb.F3. We also generated a catalytically dead mutant of the chimeric construct (nb.F3-Nedd4L[C942S]) to distinguish between ubiquitination-dependent and independent effects. Both constructs were generated in a P2A-CFP expression vector, enabling use of CFP fluorescence to confirm protein expression.

In experiments mimicking those described for nb.F3, we examined the impact of nb.F3-Nedd4L and nb.F3-Nedd4L[C942S] on reconstituted Ca$_V$2.2 channel trafficking, subunit expression levels, and whole-cell currents (*Figure 3*). Given the classical role of E3 ubiquitin ligases in mediating degradation of target proteins, we first assessed if nb.F3-Nedd4L affected total Ca$_V\beta$ expression (*Figure 3a,b*). In cells expressing BBS-$\alpha_{1B}$ + $\beta_{1b}$-YFP + $\alpha_2\delta$, neither F3-Nedd4L nor F3-Nedd4L [C942S] had any significant impact on $\beta_{1b}$ total expression as reported by the unchanged YFP fluorescence compared to negative control cells (*Figure 3a,b*). Similar results were obtained when BBS-$\alpha_{1B}$ was reconstituted with YFP-tagged $\beta_2$, $\beta_3$, or $\beta_4$ subunits, though there was a trend towards lower fluorescence with $\beta_{2a}$ and $\beta_4$ (*Figure 3b*). By contrast, nb.F3-Nedd4L significantly suppressed surface density of BBS-$\alpha_{1B}$ irrespective of the identity of the co-expressed YFP-tagged Ca$_V\beta$ (*Figure 3c*, red bars; *Figure 3—figure supplement 1*). The decreased BBS-$\alpha_{1B}$ surface density was not observed with nb.F3-Nedd4L[C942S] (*Figure 3c*, green bars), indicating it requires the catalytic activity of the attached Nedd4L HECT domain. Similarly, in cells expressing BBS-$\alpha_{1C}$ + $\beta_X$-YFP, nb.F3-Nedd4L strongly reduced Ca$_V$1.2 surface density in a ubiquitin-dependent manner (*Figure 3—figure supplement 2*).

Given the striking effect of nb.F3-Nedd4L on surface population of channels without affecting total levels Cav$\beta$, we next assessed whether there was any impact of nb.F3-Nedd4L on total $\alpha_{1B}$

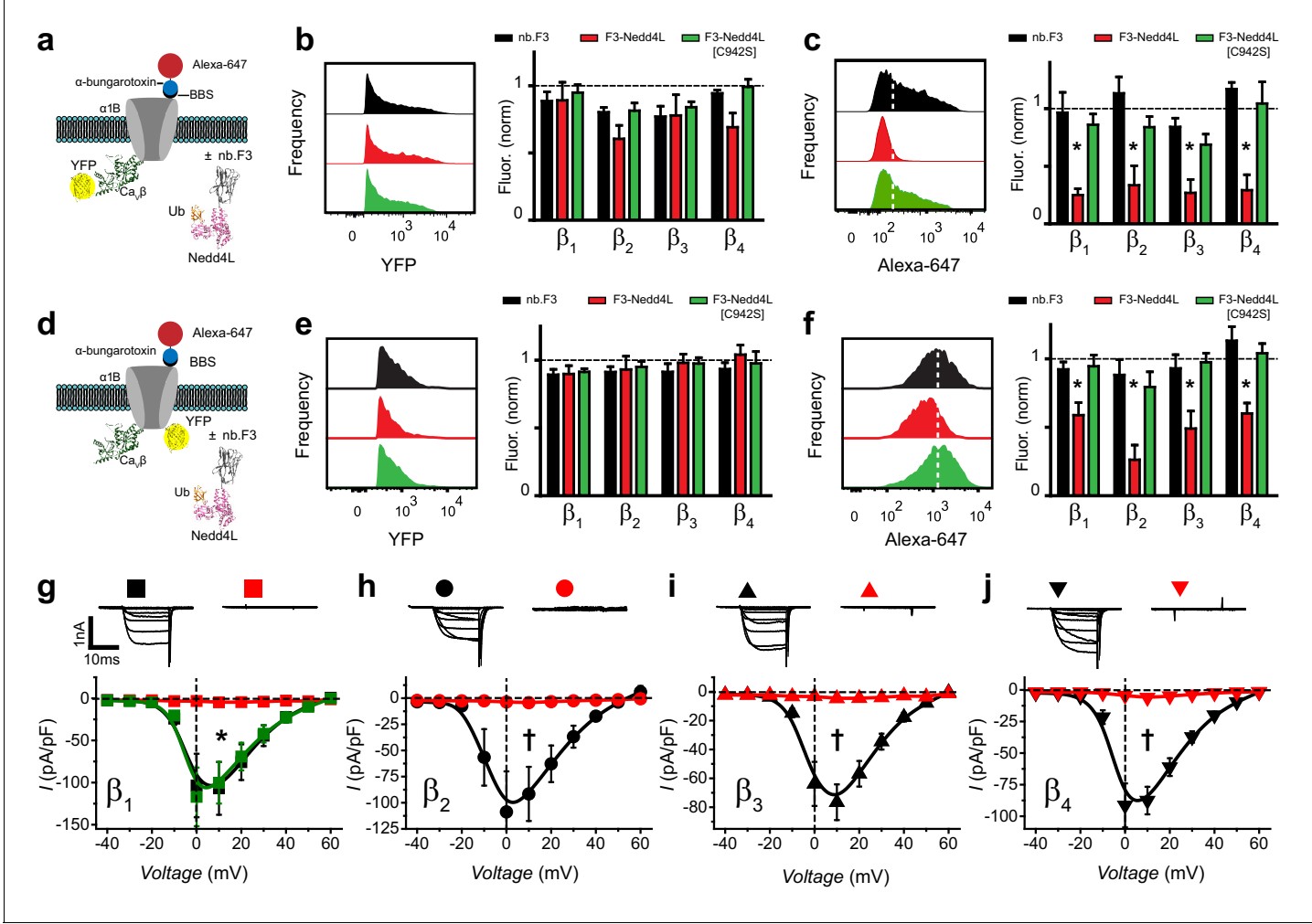

**Figure 3.** Functional impact of a chimeric nb.F3-Nedd4L protein (Ca_V-aβlator) on reconstituted Ca_v2.2 channels. (**a**) Schematic of experimental design; HEK293 cells were transfected with BBS-$\alpha_{1B}$ + Ca_Vβ-YFP + $\alpha_2\delta-1$, and either nb.F3, nb.F3-Nedd4L, or nb.F3-Nedd4L[C942S]. (**b**) Exemplar histograms (left) and summary data (right) of flow cytometry experiments measuring total (YFP) levels of Ca_Vβ_{1b}-YFP. Each data set was normalized to a control group that expressed CFP. n > 5,000 cells analyzed per experiment, N = 3 separate experiments, error bars, s.e.m. (**c**) Exemplar histograms (left) and summary data (right) of flow cytometry experiments measuring surface (647) levels of BBS-$\alpha_{1B}$. White dashed line is the threshold for 647 signal above background. (**d**) Experimental strategy; same format as in (**a**) except YFP was fused to BBS-$\alpha_{1B}$, enabling measurement of the total levels of the $\alpha_{1B}$ subunit. (**e-f**) Same format as in (**b-d**) for cells expressing BBS-$\alpha_{1B}$-YFP + Ca_Vβ + $\alpha_2\delta$-1. (**g**) Exemplar traces (top) and population *I-V* curves (bottom) from whole-cell patch clamp measurements in HEK293 cells expressing $\alpha_{1B}$ + Ca_Vβ_{1b} + $\alpha_2\delta$-1 and nb.F3 (black, $I_{peak,\ 0mV}$ = -103.5 ± 39.5 pA/pF, n=10), nb.F3-Nedd4L (red, $I_{peak,\ 0mV}$ = -3 ± 0.53 pA/pF, n=11), or nb.F3-Nedd4L[C942S] (green, $I_{peak,\ 0mV}$ = -117 ± 34.8 pA/pF, n=8). (**h-j**) Same format as (**g**) for Ca_V2.2 channels reconstituted with Ca_Vβ_2 (**h**), Ca_Vβ_3 (**i**), and Ca_Vβ_4 (**j**) with nb.F3 (black) or nb.F3-Nedd4L (red). Scale bar 1nA, 10ms. Data are means ± s.e.m., n=10 for each point. *P < 0.05 compared with control, one-way ANOVA with Tukey's multiple comparison test. †P < 0.01 compared with control, unpaired, two-tailed Student's t-test.

DOI: https://doi.org/10.7554/eLife.49253.007

The following figure supplements are available for figure 3:

**Figure supplement 1.** Exemplar flow cytometry data for Ca_V-aβlation of BBS-$\alpha_{1B}$ with YFP-Ca_Vβ_2- Ca_Vβ_4.
DOI: https://doi.org/10.7554/eLife.49253.008

**Figure supplement 2.** Functional impact of a chimeric nb.F3-Nedd4L protein (Ca_V-aβlator) on reconstituted Ca_v1.2 channels.
DOI: https://doi.org/10.7554/eLife.49253.009

subunit expression. Similar to our observations for Ca_Vβ, nb.F3-NeddL had no significant impact on the expression of BBS-$\alpha_{1B}$-YFP (*Figure 3e*, red bars) relative to either negative controls (black bars) or cells expressing nb.F3-Nedd4L[C942S] (green bars). Not surprisingly, nb.F3-Nedd4L markedly impaired surface trafficking of BBS-$\alpha_{1B}$-YFP co-expressed with any Ca_Vβ (*Figure 3f*).

Finally, we examined the functional impact of nb.F3-Nedd4L on reconstituted Ca$_V$2.2 whole-cell currents. Remarkably, nb.F3-Nedd4L essentially eliminated Ca$_V$2.2 currents reconstituted from $\alpha_{1B}$ + $\alpha_2\delta$ co-expressed with any of the four Ca$_V\beta$s (*Figure 3g-j*). Further, nb.F3-Nedd4L was equally effective in ablating whole-cell currents in reconstituted Ca$_V$1.2, Ca$_V$1.3, Ca$_V$2.1, and Ca$_V$2.3 channels (*Figure 4*).

Given its exceptional efficacy in ablating whole-cell HVACC currents via a functionalized Ca$_V\beta$-targeted nanobody, we named nb.F3-Nedd4L as Ca$_V$-a$\beta$lator, and describe the process of HVACC current elimination by this molecule as Ca$_V$-a$\beta$lation.

## Ca$_V$-a$\beta$lation of endogenous Ca$_V$1.2 channels in cardiomyocytes

We next determined whether Ca$_V$-a$\beta$lator could effectively inhibit HVACC currents in native cells where the nano-environment around Ca$_V$1/Ca$_V$2 channels is typically more complex than in heterologous cells. Cultured adult guinea pig ventricular cardiomyocytes (CAGPVCs) provided an initial

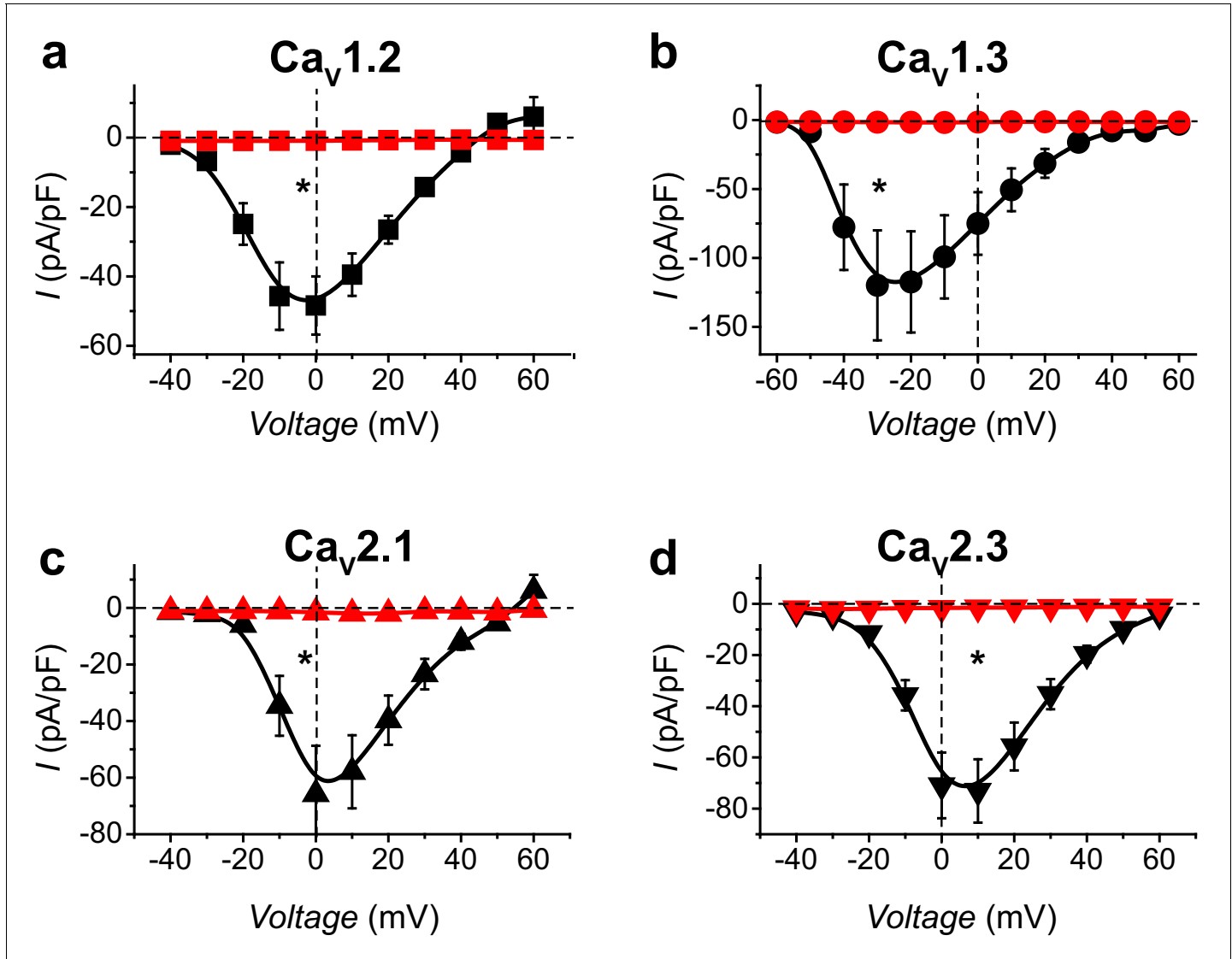

**Figure 4.** Ca$_V$-a$\beta$lator inhibits distinct reconstituted HVACCs. (a) Population I-V curves from HEK293 expressing $\alpha_{1C}$ + $\beta_{1b}$ + $\alpha_2\delta-1$ with either nb.F3 (black, $I_{peak,\ 0mV}$ = −48.4 ± 8.4 pA/pF, n = 12) or Ca$_V$-a$\beta$lator (red, $I_{peak,\ 0mV}$ = −0.93 ± 0.16 pA/pF, n = 8). (b-d) Same format as (a) for cells expressing reconstituted Ca$_V$1.3 (b), Ca$_V$2.1 (c), or Ca$_V$2.3 (d) channels. Data are means ± s.e.m. †p<0.01 compared with control, unpaired, two-tailed Student's t-test.

DOI: https://doi.org/10.7554/eLife.49253.010

exceptional challenge because they have an intricate cyto-architecture and express $Ca_V1.2$ channels that are predominantly targeted to specialized dyadic junctions. Moreover, as it has now been shown that in adult cardiomyocytes binding of $\alpha_{1C}$ to $Ca_V\beta$ is not obligatory for substantive $Ca_V1.2$ channel trafficking to the surface sarcolemma (*Yang et al., 2019*; *Meissner et al., 2011*), the fraction of $Ca_V\beta$-bound $Ca_V1.2$ channels contributing to the whole-cell L-type current ($I_{Ca,L}$) in ventricular myocytes is ambiguous. We used adenovirus to express $Ca_V$-a$\beta$lator or nb.F3-Nedd4L[C942S] in CAGPVCs which retain the rod-shaped phenotype and overall cyto-architecture of freshly isolated heart cells (*Figure 5a*). Control (non-infected) cardiomyocytes expressed $I_{Ca,L}$ that peaked at a 0 mV test pulse (*Figure 5a,b*; $I_{peak,0mV} = -6.5 \pm 0.2$ pA/pF, n = 8). By contrast, in contemporaneous experiments, cardiomyocytes expressing $Ca_V$-a$\beta$lator via adenovirus-mediated infection displayed virtually no $Ca_V1.2$ currents, demonstrating an exceptional $Ca_V$-a$\beta$lation efficiency in this system (*Figure 5a,b*; $I_{peak,0mV} = -1.0 \pm 0.3$ pA/pF, n = 9). Cardiomyocytes expressing nb.F3-Nedd4L[C942S] displayed $I_{Ca,L}$ similar to control ($I_{peak,0mV} = -5.1 \pm 0.6$ pA/pF, n = 10), indicating that ubiquitination is necessary for $Ca_V$-a$\beta$lation in this system.

What is the mechanism of $Ca_V$-a$\beta$lation in cardiomyocytes? We used immunofluorescence to probe how $Ca_V$-a$\beta$lator affected expression levels and sub-cellular localization of $Ca_V1.2$ $\alpha_{1C}$ and $\beta_2$ subunits, respectively, in cardiomyocytes. $Ca_V\alpha_{1C}$ in uninfected cardiomyocytes presented with a characteristic striated punctate distribution pattern that co-localized with that of ryanodine (RyR2) receptors (*Figure 5c*), reflecting their well-known predominant localization at dyadic junctions (*Scriven et al., 2000*; *Bers, 2002*). A similar distribution pattern for $\alpha_{1C}$ was observed in cardiomyocytes expressing nb.F3-Nedd4L[C942S], consistent with the lack of effect of this protein on $I_{Ca,L}$. In cardiomyocytes expressing $Ca_V$-a$\beta$lator, the signal intensity for punctate $\alpha_{1C}$ staining was unchanged from control cells (*Figure 5—figure supplement 1*), suggesting no impact of the presumed increase in ubiquitination on the stability of the protein. However, there was a redistribution of $\alpha_{1C}$ from dyadic junctions, as reported by a dramatic loss of co-localization between $\alpha_{1C}$ and RyR2 (*Figure 5c*). Rather, the punctate $\alpha_{1C}$ signals in $Ca_V$-a$\beta$lator-expressing cardiomyocytes coincided with Rab7, but not Rab5 or LAMP1, immunofluorescence signals (*Figure 5d*; *Figure 5—figure supplement 1*). Thus, the mechanism of $Ca_V$-a$\beta$lator inhibition of $I_{Ca,L}$ is redistribution of $\alpha_{1C}$ from dyadic junctions to intracellular compartments, specifically Rab7-positive late endosomes (*Figure 5h*) (*Rink et al., 2005*).

Cardiomyocytes expressing $Ca_V$-a$\beta$lator also showed no difference in total $Ca_V\beta_2$ levels as compared to either uninfected or nb.F3-Nedd4L[C942S]-expressing cells (*Figure 5—figure supplement 1*). Hence, $Ca_V$-a$\beta$lator-mediated redistribution of $Ca_V1.2$ in cardiomyocytes cannot be explained as simply due to an absence of $Ca_V\beta$. An intriguing possibility was that though $Ca_V$-a$\beta$lator is specifically targeted to $Ca_V\beta$ in channel complexes, it is also able to directly catalyze ubiquitination of $\alpha_1$ subunits within the macro-molecular complex. Indeed, in pulldown experiments of recombinant $Ca_V1.2$ channels, $Ca_V$-a$\beta$lator substantially increased ubiquitination of both $\alpha_{1C}$ (*Figure 5e,f*) and $Ca_V\beta_{1b}$ subunits (*Figure 5g*). Nevertheless, the overall levels of $\alpha_{1C}$ expression was unchanged with $Ca_V$-a$\beta$lator despite the increased ubiquitination (*Figure 5e*). Taken together, our results suggest that direct ubiquitination of $\alpha_{1C}$ by $Ca_V$-a$\beta$lator may underlie the redistribution of $Ca_V1.2$ channels from dyads to Rab7-positive late endosomes (*Figure 5h*).

## $Ca_V$-a$\beta$lation in dorsal root ganglion (DRG) neurons and pancreatic $\beta$ cells

We next tested the efficacy of $Ca_V$-a$\beta$lator to suppress HVACCs in murine dorsal root ganglion (DRG) neurons which were of interest because they express multiple $Ca_V1$/$Ca_V2$ channel types (*Murali et al., 2015*; *McCallum et al., 2011*), and also play a key role in the processing of noxious signals including pain and itch (*Han et al., 2013*; *Kim et al., 2016*). We infected cultured DRG neurons with adenovirus expressing either GFP, $Ca_V$-a$\beta$lator, or nb.F3-Nedd4L[C942S]. Given their heterogeneous nature, we first used fura-2 to measure calcium influx into a population of DRG neurons in response to depolarization with 40 mM KCl (*Figure 6a,b*). Recordings were done in the presence of 5 μM mibefradil to block low-voltage-activated T-type calcium channels which are also prevalent in these cells (*Puckerin et al., 2018*; *Jagodic et al., 2008*). In neurons expressing GFP or nb.F3-Nedd4L[C942S], a substantial fraction of cells displayed large increases in fura-2-reported $Ca^{2+}$ transients in response to 40 mM KCl, indicating the opening of $Ca_V1$/$Ca_V2$ channels (*Figure 6a,b*). By

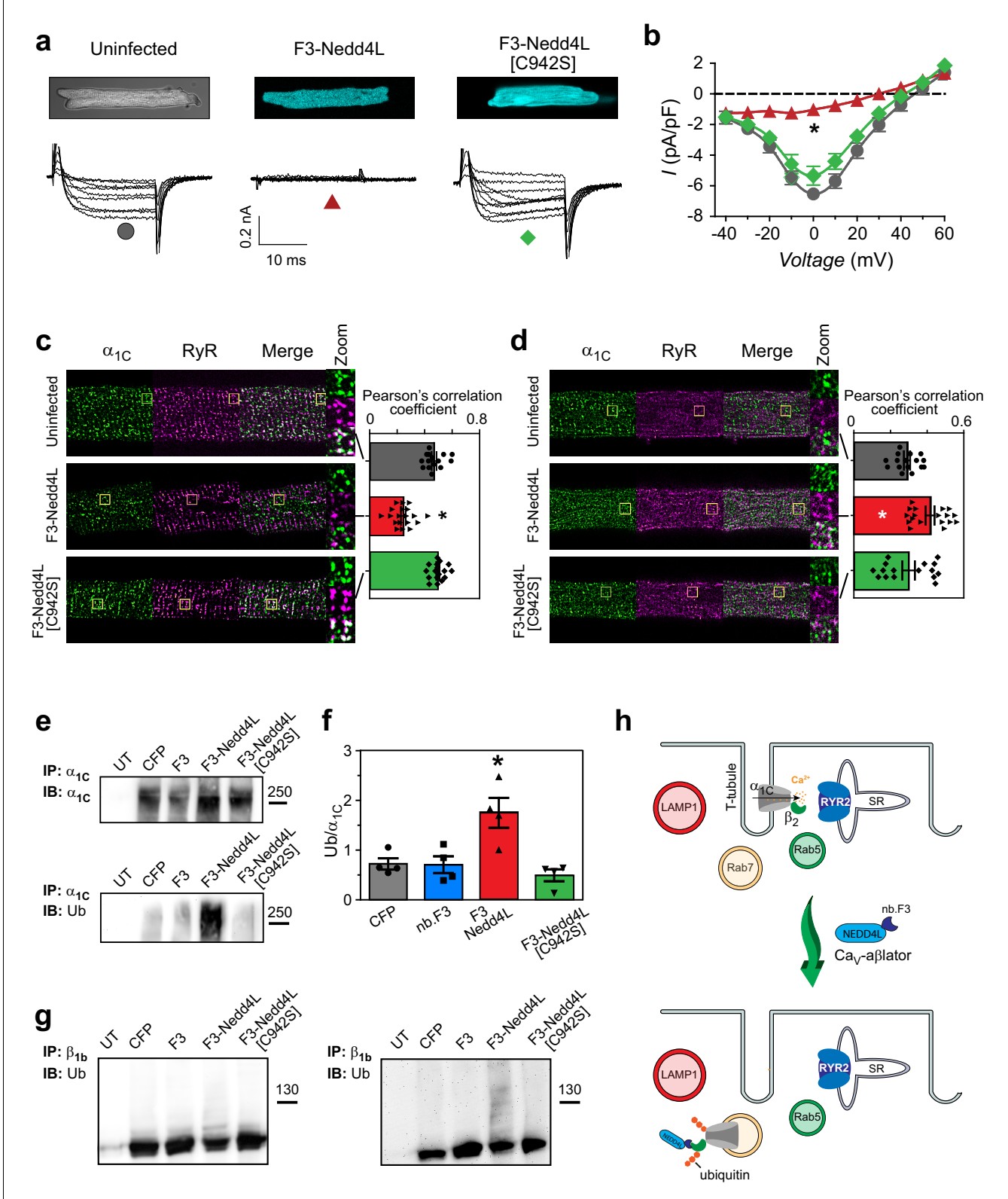

**Figure 5.** Ca$_V$-aβlation of endogenous Ca$_V$1.2 in cardiomyocytes. (a) Confocal images (top) and exemplar traces from whole-cell recordings of uninfected guinea pig cardiomyocytes (left), or infected with adenovirus expressing either Ca$_V$-βlator (middle) or nb.F3-Nedd4L[C942S] (right). Scale bar 0.2nA, 10 ms. (b) Population I-V curves from cardiomyocytes expressing Ca$_V$-βlator (red), nb.F3-Nedd4L[C942S] (green), or an uninfected control (black). (c) Left, exemplar confocal images of cardiomyocytes fixed and immunostained with α$_{1C}$ (green) and ryanodine receptor (RyR2, magenta) antibodies.
*Figure 5 continued on next page*

*Figure 5 continued*

Yellow box indicates region of high-zoom merge image.Right, co-localization between $\alpha_{1C}$ and RyR in uninfected cardiomyocytes (gray, PCC = 0.47 ± 0.02, n = 15), and those expressing either $Ca_V$-aβlator (red, PCC = 0.24 ± 0.02 n = 19), or nb.F3-Nedd4L[C942S] (green, PCC = 0.50 ± 0.01, n = 17). (**d**) Left, exemplar confocal images of fixed cardiomyocytes immunostained with $\alpha_{1C}$ (green) and Rab7 (magenta) antibodies. Yellow box indicates region of high-zoom merge image. Right, colocalization between $\alpha_{1C}$ and Rab7 in uninfected cardiomyocytes (gray, PCC = 0.29 ± 0.02, n = 16), and those expressing either $Ca_V$-aβlator (red, PCC = 0.42 ± 0.02, n = 18), or nb.F3-Nedd4L[C942S] (green, PCC = 0.30 ± 0.03, n = 16). (**e**) Pulldown of $\alpha_{1C}$ in HEK293 cells expressing $\alpha_{1C}$, $\beta_{1b}$ and either CFP, nb.F3, $Ca_V$-aβlator, or nb.F3-Nedd4L-[C942S]. Top, probing pulldown with $\alpha_{1C}$ antibody. Bottom, same blot stripped and re-probed with ubiquitin antibody. (**f**) Quantification of four separate experiments, as performed in (**e**). Data are means ± s.e.m for each point. *p<0.05 compared to control, one-way ANOVA with Tukey's multiple comparison test. (**g**) Pulldown of $Ca_V\beta_{1b}$, as in (**e**). Left, probing with $Ca_V\beta_{1b}$. Right, same blot stripped and re-probed with ubiquitin antibody. (**h**) Cartoon illustrating $Ca_V$-aβlator-induced relocation of $Ca_V1.2$ from dyads to Rab7-positive late endosomes in cardiomyocytes.

DOI: https://doi.org/10.7554/eLife.49253.011

The following figure supplement is available for figure 5:

**Figure supplement 1.** $Ca_V$-aβlator does not redistribute $\alpha_{1C}$ to Rab5 early-endosomes or lysosomes, nor decrease total levels of $\alpha_{1C}$ or $\beta_2$.

DOI: https://doi.org/10.7554/eLife.49253.012

contrast, depolarization-induced $Ca^{2+}$ influx was virtually eliminated in neurons expressing $Ca_V$-aβlator, demonstrating highly efficient $Ca_V$-ablation in this system (*Figure 6a,b*).

We used whole-cell patch clamp to further characterize the impact of $Ca_V$-aβlator on calcium currents in DRG neurons. It was of particular interest to determine relative effects of $Ca_V$-aβlator on HVACCs and LVA T-type channels that are present in a subset of DRG neurons. We recorded families of whole-cell currents evoked by test pulses (from −40 mV to +60 mV in 10 mV increments) from a holding potential of either −90 mV or −50 mV to inactivate any T-type channel present (*Figure 6c*). Cells expressing GFP (control) or F3-Nedd4L[C942S] displayed large $I_{Ba}$ irrespective of the holding potential (*Figure 6c,d*; $I_{peak,-10mV}$ = −173.9 ± 28.2 pA/pF, n = 6 for GFP, $I_{peak,-10mv}$ = −206.7 ± 36.4 pA/pF, n = 5 for F3-Nedd4L[C942S]), though those recorded with a −50 mV holding potential had a lower amplitude reflecting inactivation of T-type channels and also a fraction of HVACCs. Cells expressing $Ca_V$-aβlator displayed essentially no HVACC currents (*Figure 6c,d*; $I_{peak,-10mV}$ = −14.3 ± 6.2 pA/pF), most evident as an absence of $I_{Ba}$ recorded from a −50 mV holding potential (*Figure 6c*, middle). Moreover, in these cells, when currents were recorded from a −90 mV holding potential, they displayed fast inactivation kinetics characteristic of T-type channels (*Figure 6c*). Overall, these results indicate $Ca_V$-ablator selectively eliminates HVACCs in DRG neurons without impacting LVA T-type channels.

Finally, we tested whether $Ca_V$-ablator is also effective in murine pancreatic β-cells, which have multiple $Ca_V$ channel types ($Ca_V1.2$, $Ca_V1.3$, and $Ca_V2.1$) involved in insulin release (*Yang and Berggren, 2006*). We used adenovirus to infect digested islets isolated from transgenic mice expressing tdTomato in pancreatic β-cells. Control cells expressing GFP or nb.F3-Nedd4L[C942S] displayed robust glucose- or KCl-evoked fura-2-reported $Ca^{2+}$ transients that were essentially abolished in cells expressing $Ca_V$-aβlator (*Figure 6e-g*). Altogether, these results reveal the exceptional activity of $Ca_V$-aβlator as a genetically-encoded HVACC inhibitor that is effective across diverse cellular contexts.

## Discussion

This work introduces $Ca_V$-aβlator as a novel genetically-encoded molecule that potently inhibits HVACCs by targeting auxiliary $Ca_V\beta$ subunits. $Ca_V$-aβlator combines the exquisite specificity of a $Ca_V\beta$-targeted nanobody and the powerfully consequential catalytic activity of an E3 ubiquitin ligase. We discuss four distinct aspects of this work, based on viewing $Ca_V$-aβlator from different perspectives; 1) as a unique tool to selectively erase HVACCs in cells, 2) as a method to probe mechanisms of HVACC regulation and trafficking, 3) as a potential therapeutic, and 4) as a prototype engineered protein that enables probing new dimensions of macro-molecular membrane protein signaling.

$Ca^{2+}$ is a universal second messenger critical to the biology of virtually all cells. In excitable cells, both LVACCs and HVACCs transduce electrical signals encoded in action potentials into changes in intracellular $Ca^{2+}$ that then drive many biological responses. In cells expressing both classes of channels, the physiological effects mediated specifically through LVACCs versus HVACCs in vivo can be

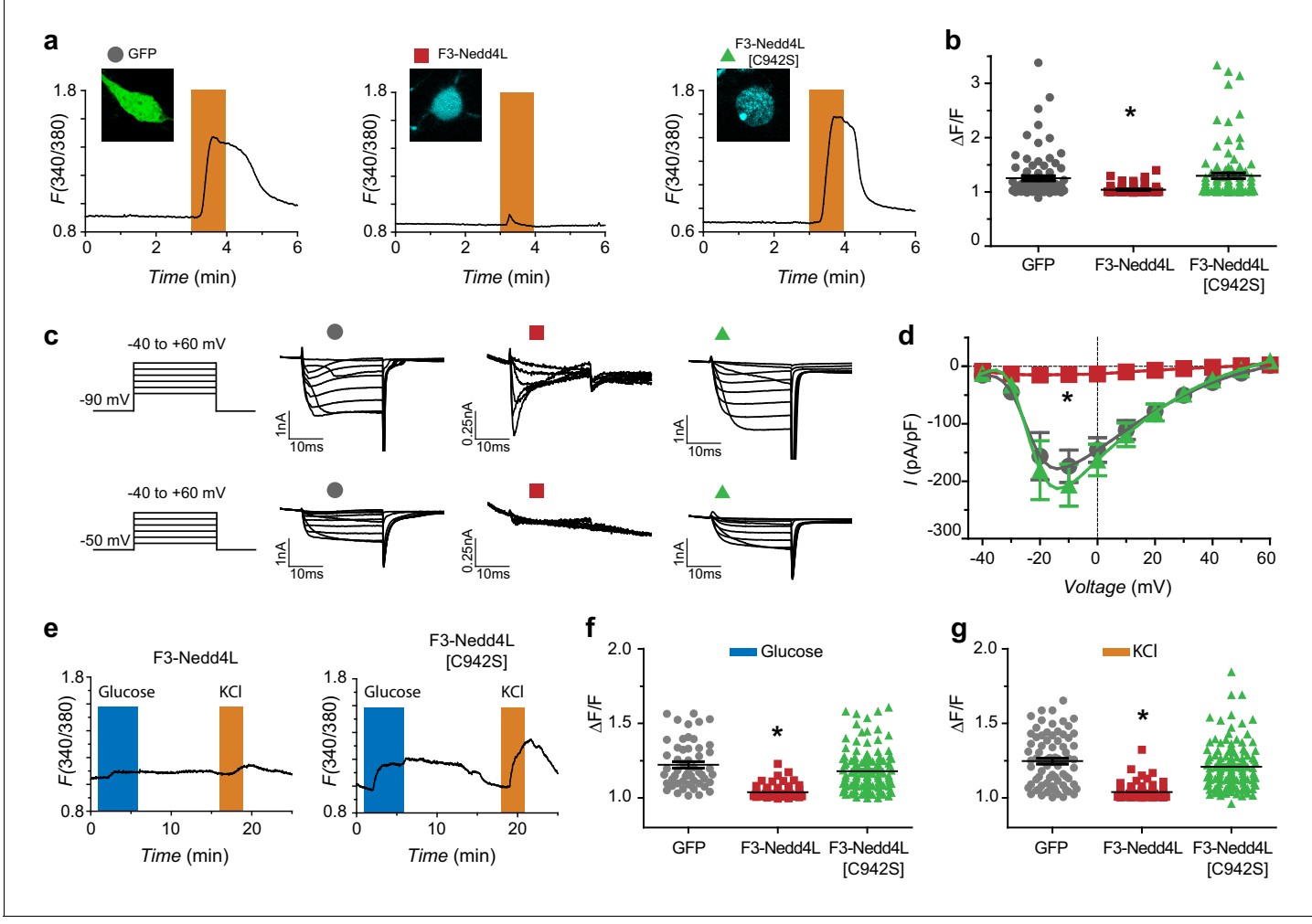

**Figure 6.** Ca$_V$-aβlation of HVACCs in DRG neurons and pancreatic β-cells. (a) Exemplar Fura-2 traces of murine DRG neurons infected with GFP (left), F3-Nedd4L (middle), F3-Nedd4L[C942S] (right), with confocal images in inset. The orange bars represent depolarization with 40 mM KCl. (b) Summary of maximum responses from neurons infected with GFP (Peak response = 1.25 ± 0.04, n = 84), F3-Nedd4L (1.04 ± 0.01, n = 77), and F3-Nedd4L[C942S] (1.30 ± 0.05, n = 92) in response to 40 mM KCl. Peaks were normalized to the baseline, defined as 1 min prior to the addition of KCl. (c) Exemplar traces of DRG neurons infected with GFP (left), F3-Nedd4L (middle), F3-Nedd4L[C942S] (right). Traces were collected at both a holding potential of −90 mV (top) and −50 mV (bottom). Notably, Ca$_V$-aβlator-infected neurons still show robust T-type current when held at −90 mV. (d) Population I-V curves from DRG neurons infected as in (a). Measurements were made at a holding potential of −90 mV. Symbols are mean currents calculated from 15 to 20 ms of a 20 ms test pulse. Data are means ± s.e.m. (e) Exemplar fura-2 traces from dispersed pancreatic islets infected with Ca$_V$-aβlator (left) or F3-Nedd4L [C942S] (right) challenged with 16.8 mM glucose (blue bars) and 40 mM KCl (orange bars). (f) Summary of maximum responses from pancreatic β-cells infected with GFP (Peak response = 1.22 ± 0.02, n = 53), F3-Nedd4L (Peak response = 1.04 ± 0.01, n = 62), and F3-Nedd4L[C942S] (Peak response = 1.18 ± 0.01, n = 122) in response to 16.8 mM glucose. (g) Summary of maximum responses from pancreatic β-cells infected with GFP (Peak response = 1.25 ± 0.02, n = 77), F3-Nedd4L (Peak response = 1.04 ± 0.01, n = 62), and F3-Nedd4L[C942S] (Peak response = 1.21 ± 0.01, n = 122) in response to 40 mM KCl. Data are means ± s.e.m. *p<0.05 compared to control, one-way ANOVA with Tukey's multiple comparison test.
DOI: https://doi.org/10.7554/eLife.49253.013

difficult to decipher. Ca$_V$-aβlator now presents as a tool that can be deployed in target cells to virtually erase all HVACCs while leaving LVACC actions intact. The closest existing proteins that can similarly eliminate HVACCs are RGK GTPases which are capable of potently inhibiting Ca$_V$1/Ca$_V$2 channels when over-expressed in target cells (*Murata et al., 2004*; *Chen et al., 2005*; *Xu et al., 2010*; *Puckerin et al., 2018*; *Bannister et al., 2008*). However, a distinct disadvantage of RGKs is their propensity for off-target effects due to their known interactions with, and regulation of, cytoskeletal proteins and other signaling molecules including 14-3-3, calmodulin, and CaM kinase II (*Yang and Colecraft, 2013*; *Correll et al., 2008*; *Royer et al., 2018*; *Béguin et al., 2005*). Over the

last two decades, several groups have sought to disrupt the $\alpha_1$-Ca$_V\beta$ interaction with either small molecules or by over-expressing the AID peptide as a Ca$_V\beta$ sponge (*Findeisen et al., 2017*; *Chen, 2018*; *Khanna et al., 2019*; *Yang et al., 2019*). While this approach has shown some efficacy in certain instances, the potency of HVACC inhibition falls well short of that achieved here with Ca$_V$-aβlator. Indeed, over-expressing the AID peptide in adult cardiac myocytes is not effective in inhibiting Ca$_V$1.2 channels (*Yang et al., 2019*), because in this context $\alpha_{1C}$ binding to Ca$_V\beta$ is not absolutely required for channel trafficking to the surface (*Yang et al., 2019*; *Meissner et al., 2011*). Nevertheless, the ability of Ca$_V$-aβlator to essentially eradicate $I_{Ca,L}$ in adult cardiomyocytes indicates that under normal physiological conditions essentially all $\alpha_{1C}$ subunits are associated with a Ca$_V\beta$ in ventricular heart cells.

Ca$_V$1/Ca$_V$2 channels and other surface membrane proteins spend a significant portion of their life cycles in intracellular compartments reflecting their biogenesis, recycling, and ultimate destruction. The signals regulating HVACC degradation and trafficking among compartments are arcane and poorly understood, but likely prominently involve post-translational modifications of channel subunits. Here, we show that targeted ubiquitination of $\alpha_{1C}/\beta_2$ complexes in cardiomyocytes with Ca$_V$-aβlator specifically arrests Ca$_V$1.2 channels in Rab7-positive late endosomes. Ca$_V$-aβlator possesses the catalytic HECT domain of Nedd4L which is known to principally catalyze the addition of K63-linkage polyubiquitin chains to target proteins (*Kim and Huibregtse, 2009*; *Scheffner and Kumar, 2014*). Thus, our results suggest that K63-ubiquitin chains on $\alpha_{1C}/\beta_2$ subunits may be a key signal directing Ca$_V$1.2 channels to late endosomes. We further found that targeted ubiquitination of HVACC $\alpha_1$ subunits with Ca$_V$-aβlator did not lead to their enhanced degradation either in heterologous cells or cardiomyocytes. By contrast, using a GFP nanobody to target the Nedd4L HECT domain to YFP-tagged KCNQ1, a known substrate of endogenous Nedd4L, resulted in reduced expression of this K$^+$ channel pore-forming $\alpha_1$ subunit (*Kanner et al., 2017*). Hence, the impact of Nedd4L HECT domain on the stability of membrane proteins is likely substrate-dependent. We speculate that arming nb.F3 with the catalytic domains of other types of E3 ligases that catalyze formation of different polyubiquitin chains will elucidate the precise signals dictating Ca$_V$1/Ca$_V$2 channel degradation and trafficking among distinct compartments. Beyond ubiquitination, the approach could also be potentially used to elucidate functional consequences and mechanisms of other post-translational modifications such as phosphorylation/dephosphorylation on Ca$_V$1/Ca$_V$2 channels, as well as to localize sensors that report on signals within HVACC nano-domains in live cells.

Blocking the activity of specific HVACCs with small molecules is a prevailing or potential therapy for many cardiovascular and neurological diseases including; pain, hypertension, cardiac arrhythmias, epilepsy, and Parkinson's disease (*Zamponi, 2016*). A limitation of small molecule or toxin blockers for HVACCs is the propensity for off-target effects due to their inevitable widespread distribution when administered to a patient. In some circumstances such off-target effects may limit the therapeutic window sufficiently to adversely affect treatment efficacy. Genetically-encoded HVACC inhibitors have great potential to be useful therapeutics with the advantage that their expression can be restricted to target tissues/cell types, or even to spatially discrete channels within single cells (*Murata et al., 2004*; *Makarewich et al., 2012*). Given its potency in silencing HVACC activity, Ca$_V$-aβlator could be a lead molecule for future development into a gene therapy for particular applications where a genetically-encoded HVACC inhibitor is warranted. For this purpose, it may be desirable to generate Ca$_V$-aβlator versions whose time course and extent of action could be tuned by either a small molecule or light. Indeed, this is a focus of ongoing work.

Finally, an exciting prospect is the potential of Ca$_V$-aβlator as a prototype that can be further developed to engineer proteins that regulate Ca$_V$1/Ca$_V$2 channel complexes with new dimensions of specificity. For example, a prevailing idea is that Ca$_V$1/Ca$_V$2 channels of a particular type (e.g. Ca$_V$1.2 channels in cardiomyocytes) may yet form discrete signaling units with different functional outputs in single cells based on their incorporation into divergent macro-molecular complexes (*Shaw and Colecraft, 2013*). There are tantalizing hints that different Ca$_V\beta$ isoforms could be a node of signal diversification by promoting formation of molecularly distinct HVACC macro-molecular complexes (*McEnery et al., 1998*; *Brice and Dolphin, 1999*; *Campiglio and Flucher, 2015*). Hence, the ability to inhibit specific Ca$_V$ channel macro-molecular complexes based on the identity of the constituent Ca$_V\beta$ is biologically important, yet not rigorously addressable with conventional knockout/knockdown approaches. However, this capability may be readily achieved with Ca$_V$-aβlators directed towards particular Ca$_V\beta$ isoforms. A challenge to realize this possibility is the

development of Ca$_V$β isoform-specific nanobodies which should be feasible given that there is sequence divergence among Ca$_V$βs outside the conserved *src* homology 3 (SH3) and guanylate kinase (GK) domains (*Buraei and Yang, 2010*). In a broader context, the phenomenon of ion channel pore-forming α$_1$ subunits assembled with diverse auxiliary subunits in individual cells is common throughout biology (*O'Malley and Isom, 2015*; *Copits and Swanson, 2012*; *Trimmer, 2015*). Hence, Ca$_V$-aβlator-inspired molecules and approaches might be expected to elucidate functional dimensions of ion channel macro-molecular complex signaling that, to date, have remained refractory to analyses.

# Materials and methods

## Key resources table

| Reagent type (species) or resource | Designation | Source or reference | Identifiers | Additional information |
|---|---|---|---|---|
| Gene (rat) | CACNA1B | | NM_147141 | |
| Gene (rabbit) | CACNA1C | | NM_001136522 | |
| Gene (rat) | CACNB1 | | NM_017346 | |
| Gene (human) | CACNB2 | | NM_201590 | |
| Gene (rat) | CACNB3 | | NM_012828.2 | |
| Gene (rat) | CACNB4 | | NM_001105733.1 | |
| Gene (human) | CACNA2D1 | | NM_000722.4 | |
| Gene (human) | NEDD4L | | NM_001144965.2 | |
| strain, strain background (*Escherichia coli*) | Rosetta DE3 | Millipore Sigma | | |
| Cell line (Human) | HEK293 | Other | RRID: CVCL_0045 | Laboratory of Dr. Robert Kass |
| Recombinant DNA reagent | nb.F3-CFP-PKC$_\gamma$ | This paper | | Made by PCR, see molecular biology and cloning |
| Recombinant DNA reagent | nb.F3-P2A-CFP | This paper | | Made by PCR, see molecular biology and cloning |
| Recombinant DNA reagent | nb.F3-Nedd4L-P2A-CFP | This paper | | Made from pCI HA Nedd4L (Addgene #27000); see molecular biology and cloning |
| Recombinant DNA reagent | nb.F3-Nedd4L [C942S]-P2A-CFP | This paper | | Made by site-directed mutagenesis |
| Recombinant DNA reagent | BBS-α$_{1B}$ | PMID: 20308247 | | |
| Recombinant DNA reagent | BBS-α$_{1C}$ | PMID: 20308247 | | |
| Antibody | Anti-α$_{1C}$ | Alomone | Cat#: ACC-003 | 1:1000 WB/IF |

*Continued on next page*

Continued

| Reagent type (species) or resource | Designation | Source or reference | Identifiers | Additional information |
|---|---|---|---|---|
| Antibody | Anti-$\alpha_{1C}$ | NeuroMab | Clone: N263/31 | 1:200 IF |
| Antibody | Anti-$Ca_V\beta_1$ | NeuroMab | Clone: N7/18 | 1:500 WB |
| Antibody | Anti-$Ca_V\beta_2$ | Alomone | Cat#: ACC-105 | 1:200 |
| Antibody | Anti-Rab5 | Cell Signaling Technology | Cat#: 3547 | 1:200 |
| Antibody | Anti-Rab7 | Cell Signaling Technology | Cat#: 9367 | 1:200 |
| Antibody | Anti-LAMP1 | Developmental Studies Hybridoma Bank at the University of Iowa | RRID: AB_528127 | 1:100 |
| Antibody | Anti-RyR | Thermo Fisher Scientific | Cat#: MA3-916 | 1:1000 |
| Antibody | Anti-actin | Sigma | Cat#: A5060 | 1:1000 |
| Antibody | Anti-ubiquitin, VU-1 | LifeSensors | Cat#: VU101 | 1:500 |
| Antibody | RFP-trap agarose beads | Chromotek | Cat#: rta-20 | |
| Antibody | Anti-FLAG affinity gel | Sigma-Aldrich | Cat#: A2220 | |
| Peptide, recombinant reagent | FLAG peptide | Sigma-Aldrich | Cat#: F3290 | |
| Peptide, recombinant reagent | Ni-NTA agarose | Qiagen | Cat#: 30210 | |
| Peptide, recombinant reagent | Protein A/G sepharose beads | Rockland | | |
| Peptide, recombinant reagent | $\alpha$-bungarotoxin, Alexa Fluor 647 conjugate | Life Technologies | | |
| Peptide, recombinant reagent | Fura-2 AM | Life Technologies | Cat#: F1221 | |
| chemical compound, drug | Phorbol 12,13-dibutyrate | Sigma-Aldrich | Cat#: P1269 | |
| commercial assay or kit | AdEasy Adenoviral Vector Systems | Stratagene | | |
| commercial assay or kit | QuikChange Lightning Site-Directed Mutagenesis Kit | Stratagene | | |
| software, algorithm | FlowJo | | RRID: SCR_008520 | |
| software, algorithm | PulseFit | HEKA | | |
| software, algorithm | EasyRatioPro | HORIBA | | |
| software, algorithm | GraphPad Prism | | RRID: SCR_002798 | |

## Protein purification

We used the BacMam expression system to purify $Ca_V\beta_{1B}$ and $Ca_V\beta_3$ (*Goehring et al., 2014*). Briefly, full-length $Ca_V\beta_{1b}$ and $Ca_V\beta_3$ were cloned into a modified pEG BacMam vector with a C-terminal FLAG tag using BamHI and EcoRI sites. BacMam virus was subsequently generated in Sf9 cells and harvested after three rounds of amplification. 100 mL of BacMam virus was used to infect 1 L of HEK293 GnTI⁻ cells (N-acetylglucosaminyltransferase I-negative) and kept shaking at 37°C. After 18 hrs the cells were stimulated with 10 mM sodium butyrate and harvested 72 hrs later. Cells were lysed using an Avestin Emulsiflex-C3 homogenizer in buffer containing 50 mM Tris, 150 mM KCl, 10% sucrose, 1 mM PMSF (phenylmethylsulfonyl fluoride), and EDTA-free Complete protease inhibitor cocktail (Roche), pH 7.4. Lysate was spun down at 35,000 g for 1 hr. $Ca_V\beta$ was subsequently isolated from supernatant with anti-FLAG antibody (M2) affinity chromatography, and eluted with 100 µg/mL FLAG peptide (Sigma Millipore) in 50 mM TrisHCl, 150 mM KCl, pH 7.4. The protein was then applied to an ion exchange column (MonoQ, GE) and eluted with a linear KCl gradient of 50 mM to 1M. Peak fractions were collected and subjected to size exclusion chromatography (Superdex 200, GE) in a buffer containing 20 mM Tris, 150 mM KCl, pH 7.4. Proteins were brought to 20% glycerol, flash frozen, and stored at −80°C.

For isothermal titration calorimetry experiments, both $Ca_V\beta_{2b}$ and nb.F3 were cloned via Gibson assembly (*Gibson et al., 2009*) into an IPTG (isopropyl β-D-1-thiogalactopyranoside) inducible, kanamycin-resistant pET derived plasmid (Novagen, Madison, Wisconsin), with an N-terminal deca-histidine tag (His10) and transformed into Rosetta DE3 *E. coli* (Millipore Sigma), following manufacturers' instructions. Cells were grown at 37°C in 1L 2xTY media supplemented with 50 ug/mL carbenicillin and 35 µg/mL chloramphenicol and shook at 225 rpm. Protein expression was induced with 0.2 mM IPTG when the cells reached an OD of 0.6–0.8. The cells were then grown overnight at 22°C.

Nb.F3 was purified as previously described (*McMahon et al., 2018*): briefly, cells were harvested and resuspended in 100 mL buffer containing (mM) 500 sucrose, 200 Tris (pH 8), 0.5 EDTA and osmotically shocked with the addition of 200 mL water with stirring. The lysate was brought to a concentration of (mM) 150 NaCl, 2 MgCl₂, and 20 imidazole and centrifuged at 20,000 g, 4°C for 30 min. The supernatant was combined with 2 mL Ni-NTA Sepharose resin (Qiagen) in batch, washed with 70 mM imidazole, and eluted with 350 mM imidazole. The eluant was dialyzed into a buffer containing 150 mM NaCl, 10 mM HEPES, pH 7.4 and purified with an S200 size exclusion column (GE Healthcare).

For the purification of $Ca_V\beta_{2b}$, cells were pelleted and resuspended in a buffer containing (mM) 300 NaCl, 20 Tris HCl, 10% glycerol, pH 7.4, 0.5 PMSF, and EDTA-free Complete protease inhibitor cocktail (Roche). Cells were lysed using an Avestin Emulisflex-C3 homogenizer and spun at 35,000 g for 30'. The solubilized protein was applied to Ni-NTA Sepharose (Qiagen) and purified as nb.F3.

## Nanobody generation

One llama was immunized with an initial injection of 600 µg purified $Ca_V\beta_{1b}$ and $Ca_V\beta_3$, with four boosters of 200 ug each protein administered every other week (Capralogics Inc, Hardwick, MA). 87 days after the first immunization, lymphocytes were isolated from blood and a cDNA library with ProtoScript II Reverse Transcriptase (New England Biolabs). Nanobodies were isolated as previously described (*Pardon et al., 2014*), using a two-step nested PCR. Amplified Vhh genes were cloned into the phagemid plasmid pComb3xSS, a gift from Carlos Barbas (*Andris-Widhopf et al., 2000*) (Addgene plasmid # 63890). A phage display library was created using electrocompetent TG1 *E. coli* cells (Lucigen). Three rounds of phage display were performed as previously described (*Pardon et al., 2014*), using 100 nM biotinylated $Ca_V\beta_3$ as bait on neutravidin-coated Nunc-Immuno plates (Thermo Scientific). Clones of interest were subsequently cloned into mammalian expression systems for further study (see below).

## Isothermal titration calorimetry

Isothermal Titraction calorimetry measurements were performed using an MicroCal Auto iTC 200 (Malvern Panalytical) at 25°C. Samples were dialyzed into 300 mM NaCl, 20 mM HEPES, 5% glycerol, pH 7.5 and filtered beforehand. Injections of 2 µL nb.F3 into 400 µL of $Ca_V\beta_{2b}$. Data were processed with MicroCal Origin 7.0.

## Molecular biology and plasmid construction

Potential nbs were PCR amplified with primers flanking their conserved framework (FW) FW1 and FW4 regions and inserted into the mammalian expression plasmid pcDNA3 (Invitrogen) using HindIII and EcoRI sites. An additional GSG linker was included in the PCR and the insert was ligated upstream of an enhanced CFP and C1 domain of human PKCγ (residues 51–180).

Rat $Ca_V\beta_{1b}$, a kind gift from Dr. Jian Yang (Columbia University), was PCR amplified for subsequent overlap PCR with YFP, inserting a GSG linker between the two proteins. The resulting $Ca_V\beta_{1b}$-GSG-YFP sequence was digested with BamHI and NotI and ligated into a PiggyBac CMV mammalian expression vector (System Biosciences). A similar cloning strategy was used for $Ca_V\beta_3$ and $Ca_V\beta_4$. Rat $Ca_V\beta_{2a}$ was PCR amplified with an N-terminal YFP to prevent palmitoylation of the $\beta_{2a}$ subunit (*Chien et al., 1996*) and inserted with a similar strategy.

A customized bicistronic vector (xx-P2A-CFP) was synthesized in the pUC57 vector, in which coding sequence for P2A peptide was sandwiched between an upstream multiple cloning site and enhanced cyan fluorescent protein (CFP) (Genewiz). The xx-P2A-CFP fragment was amplified by PCR and cloned into the PiggyBac CMV mammalian expression vector (System Biosciences) using NheI/NotI sites. To generate nb.F3 -P2A-CFP, we PCR amplified the coding sequence for nb.F3 and cloned it into xx-P2A-CFP using NheI/AflII sites. A similar backbone was created in the PiggyBac CMV mammalian expression vector in which CFP-P2A-xx contained a multiple cloning site downstream of the P2A site (Genewiz). Nb.F3 was PCR amplified and ligated into the vector with BglII/AscI sites. The HECT domain of human Nedd4L (*Gao et al., 2009*) (a gift from Joan Massague, Addgene plasmid # 27000) consisting of residues 594–974 was PCR amplified and inserted downstream of nb.F3 using AscI/AgeI sites. Mutagenesis of C942S was accomplished using site-directed mutagenesis.

$\alpha_{1B}$-BBS, harboring two tandem 13 residue bungarotoxin-binding sites (SWRYYESSLEPYPD) in the domain IV S5-S6 extracellular loop, was a kind gift from Dr. Steven Ikeda (NIAAA). $\alpha_{1C}$ and $\alpha_{1C}$-BBS, and $\alpha_{1C}$-BBS-YFP have been described previously (*Yang et al., 2010*; *Kanner et al., 2017*).

## Cell culture and transfection

Human embryonic kidney (HEK293) cells were a kind gift from the laboratory of Dr. Robert Kass (Columbia University). Cells were mycoplasma free, as determined by the MycoFluor Mycoplasma Detection Kit (Invitrogen, Carlsbad, CA). Low passage HEK293 cells were cultured at 37°C in DMEM supplemented with 5% fetal bovine serum (FBS) and 100 mg/mL of penicillin–streptomycin. HEK293 cell transfection was accomplished using the calcium phosphate precipitation method. Briefly, plasmid DNA was mixed with 7.75 µL of 2 M $CaCl_2$ and sterile deionized water (to a final volume of 62 µL). The mixture was added dropwise, with constant tapping to 62 µL of 2x Hepes buffered saline containing (in mM): Hepes 50, NaCl 280, $Na_2HPO_4$ 1.5, pH 7.09. The resulting DNA–calcium phosphate mixture was incubated for 20 min at room temperature and then added dropwise to HEK293 cells (60–80% confluent). Cells were washed with $Ca^{2+}$-free phosphate buffered saline after 4–6 hr and maintained in supplemented DMEM.

Isolation of adult guinea pig cardiomyocytes was performed in accordance with the guidelines of Columbia University Animal Care and Use Committee. Prior to isolation, plating dishes were precoated with 15 µg/mL laminin (Gibco). Adult female Hartley guinea pigs (Charles River) were euthanized with 5% isoflurane, hearts were excised and ventricular myocytes isolated by first perfusing in KH solution (mM): 118 NaCl, 4.8 KCl, 1 $CaCl_2$ 25 HEPES, 1.25 $K_2HPO_4$, 1.25 $MgSO_4$, 11 glucose,. 02 EGTA, pH 7.4, followed by KH solution without calcium using a Langendorff perfusion apparatus. Enzymatic digestion with 0.3 mg/mL Collagenase Type 4 (Worthington) with 0.08 mg/mL protease and. 05% BSA was performed in KH buffer without calcium for six minutes. After digestion, 40 mL of a high $K^+$ solution was perfused through the heart (mM): 120 potassium glutamate, 25 KCl, 10 HEPES, 1 $MgCl_2$, and. 02 EGTA, pH 7.4. Cells were subsequently dispersed in high $K^+$ solution. Healthy rod-shaped myocytes were cultured in Medium 199 (Life Technologies) supplemented with (mM): 10 HEPES (Gibco), 1x MEM non-essential amino acids (Gibco), 2 L-glutamine (Gibco), 20 D-glucose (Sigma Aldrich), 1% vol vol$^{-1}$ penicillin-streptomycin-glutamine (Fisher Scientific),. 02 mg/mL Vitamin B-12 (Sigma Aldrich) and 5% (vol/vol) FBS (Life Technologies) to promote attachment to dishes. After 5 hr, the culture medium was switched to Medium 199 with 1% (vol/vol) serum, but

otherwise supplemented as described above. Cultures were maintained in humidified incubators at 37°C and 5% $CO_2$.

Murine dorsal root ganglion (DRG) neurons were kindly provided by the laboratory of Dr. Ellen Lumpkin (Columbia University). DRG neurons were isolated as previously described (*Albuquerque et al., 2009*). DRG neurons were plated onto glass coverslips coated with 15 µg/mL laminin (Corning) and maintained in Neurobasal media (Thermo Fisher Scientific) supplemented with 1x B-27 (Thermo Fisher Scientific), 100 µg mL$^{-1}$ penicillin/streptomycin (Fisher Scientific), 0.29 mg/mL L-glutamine (Gibco), 50 ng mL$^{-1}$ NGF (Sigma Aldrich), 2 ng mL$^{-1}$ GDNF (Sigma Aldrich), and 10 µM cytosine β-D-arabinofuranoside (Sigma Aldrich).

## Pancreatic beta cell isolation and culture

Murine pancreatic β-cells from Rip-Cre (Jackson Laboratories Stock #003573) mice crossed with Rosa26-tdTomato (Jackson Laboratories Stock #007909) mice were kindly provided by the laboratory of Dr. Domenico Accili (Columbia University). Islets were isolated as previously described (*Stull et al., 2012*), dispersed with 0.05% trypsin EDTA (Gibco) and plated onto 35 mm glass bottom dishes with 10 mm microwells (Cellvis) pre-coated with 10 mg/mL fibronectin (Sigma Aldrich). Islets were maintained in RPMI 1640 media (Corning) supplemented with 15% FBS and 100 µg mL$^{-1}$ penicillin/streptomycin. Islets were imaged 24–48 hr after adenoviral infection.

## Adenoviral generation

Adenoviral vectors expressing GFP and CFP-P2A-nb.F3-Nedd4L[C942S] were generated using the pAdEasy system (Stratagene) according to manufacturer's instructions as previously described (*Kanner et al., 2017*; *Subramanyam, 2013*). Plasmid shuttle vectors (pShuttle CMV) containing cDNA for CFP-P2A-nb.F3-Nedd4L[C942S] were linearized with PmeI and electroporated into BJ5183-AD-1 electrocompetent cells pre-transformed with the pAdEasy-1 viral plasmid (Stratagene). PacI restriction digestion was used to identify transformants with successful recombination. Positive recombinants were amplified using XL-10-Gold bacteria, and the recombinant adenoviral plasmid DNA linearized with PacI digestion. HEK cells cultured in 60 mm diameter dishes at 70–80% confluency were transfected with PacI-digested linearized adenoviral DNA. Transfected plates were monitored for cytopathic effects (CPEs) and adenoviral plaques. Cells were harvested and subjected to three consecutive freeze-thaw cycles, followed by centrifugation (2,500 × g) to remove cellular debris. The supernatant (2 mL) was used to infect a 10 cm dish of 90% confluent HEK293 cells. Following observation of CPEs after 2–3 d, cell supernatants were used to re-infect a new plate of HEK293 cells. Viral expansion and purification was carried out as previously described (*Colecraft et al., 2002*). Briefly, confluent HEK293 cells grown on 15 cm culture dishes (x8) were infected with viral supernatant (1 mL) obtained as described above. After 48 hr, cells from all of the plates were harvested, pelleted by centrifugation, and resuspended in 8 mL of buffer containing (in mM) 20 Tris HCl, 1 $CaCl_2$, one and $MgCl_2$ (pH 8). Cells were lysed by four consecutive freeze-thaw cycles and cellular debris pelleted by centrifugation. The virus-laden supernatant was purified on a cesium chloride (CsCl) discontinuous gradient by layering three densities of CsCl (1.25, 1.33, and 1.45 g/mL). After centrifugation (50,000 rpm; SW41Ti Rotor, Beckman-Coulter Optima L-100K ultracentrifuge; 1 hr, 4°C), a band of virus at the interface between the 1.33 and 1.45 g/mL layers was removed and dialyzed against PBS (12 hr, 4°C). Adenoviral vector aliquots were frozen in 10% glycerol at −80°C until use. Generation of CFP-P2A-nb.F3-Nedd4L was performed by Vector Biolabs (Malvern, PA).

## Flow cytometry assay of total and surface calcium channels

Cell surface and total ion channel pools were assayed by flow cytometry in live, transfected HEK293 cells as previously described (*Kanner et al., 2017*; *Aromolaran et al., 2014*). Briefly, 48 hr post-transfection, cells cultured in 12-well plates were gently washed with ice cold PBS containing $Ca^{2+}$ and $Mg^{2+}$ (in mM: 0.9 $CaCl_2$, 0.49 $MgCl_2$, pH 7.4), and then incubated for 30 min in blocking medium (DMEM with 3% BSA) at 4°C. HEK293 cells were then incubated with 1 µM Alexa Fluor 647 conjugated α-bungarotoxin (BTX$_{647}$; Life Technologies) in DMEM/3% BSA on a rocker at 4°C for 1 hr, followed by washing three times with PBS (containing $Ca^{2+}$ and $Mg^{2+}$). Cells were gently harvested in $Ca^{2+}$-free PBS, and assayed by flow cytometry using a BD Fortessa Cell Analyzer (BD

Biosciences, San Jose, CA, USA). CFP- and YFP-tagged proteins were excited at 407 and 488 nm, respectively, and Alexa Fluor 647 was excited at 633 nm.

## Electrophysiology

Whole-cell recordings of HEK293 cells were conducted 48 hr after transfection using an EPC-10 patch clamp amplifier (HEKA Electronics) controlled by Pulse software (HEKA). Micropipettes were prepared from 1.5 mm thin-walled glass (World Precision Instruments) using a P97 microelectrode puller (Sutter Instruments). Internal solution contained (mM): 135 cesium-methansulfonate (CsMeSO$_3$), 5 CsCl, 5 EGTA, 1 MgCl$_2$, 2 MgATP, and 10 HEPES (pH 7.3). Series resistance was typically between 1–2 MΩ. There was no electronic resistance compensation. External solution contained (mM): 140 tetraethylammonium-MeSO$_3$, 5 BaCl$_2$, and 10 HEPES (pH 7.4). Whole-cell I-V curves were generated from a family of step depolarizations (−60 mV to +80 mV from a holding potential of −90 mV). Currents were sampled at 20 kHz and filtered at 5 kHz. Traces were acquired at a repetition interval of 10 s. Leak and capacitive transients were subtracted using a P/4 protocol.

Whole-cell recordings of cardiomyocytes and DRG neurons were performed 48 hr after infection. HEK cell internal and external solutions were used for DRG experiments. Whole-cell recordings for guinea pig cardiomyocytes used internal solution comprised of (mM): 150 CsMeSO$_3$, 10 EGTA, 5 CsCl, MgCl$_2$, 4 MgATP, and 10 HEPES. For formation of gigaohm seals and initial break-in to the whole-cell configuration, cells were perfused in Tyrode solution containing (mM): 138 NaCl, 4 KCl, 2 CaCl$_2$, 1 MgCl$_2$, 0.33 NaH$_2$PO$_4$, and 10 HEPES (pH 7.4). Upon successful break-in, the perfusing media was switched to an external solution composed of (mM): 155 N-methyl-D-glucamine, 10 4-amino-pyridine, 1 MgCl$_2$, 5 BaCl$_2$, and 10 HEPES (pH 7.4). Currents were sampled at 20 kHz and filtered at 5 kHz. Leak and capacitive transients were subtracted using a P/4 protocol.

## Immunofluorescence staining

Approximately 48 hr after adenoviral infection, guinea pig cardiomyocytes were fixed in 4% paraformaldehyde (wt/vol, in PBS) for 20 min at RT. Cells were washed twice with PBS and then incubated in 0.1M glycine (in PBS) for 10 min at RT to block free aldehyde groups. Fixed cells were then permeabilized with 0.2% Triton X-100 (in PBS) for 20 min at RT. Non-specific binding was blocked with a 1 hr incubation at RT in PBS solution containing 3% (vol vol$^{-1}$) normal goat serum (NGS), 1% BSA, and 0.1% Triton X-100. Cells were then incubated with primary antibody in PBS containing 1% NGS, 1% BSA, and 0.1% BSA overnight at 4°C. Cells were washed three times for 10 min each with PBS with 0.1% Triton X-100 and then stained with secondary antibody for 1 hr at RT. Antibody dilutions were prepared in PBS solution containing 1% NGS, 1% BSA, and 0.1% Triton X-100. The cells were then washed in PBS with 0.1% Triton X-100 and imaged in the same solution. Primary antibodies and working dilutions were as follows: $\alpha_{1C}$: Alomone, 1:1000; UC Davis/NIH NeuroMab Facility, clone N263/31, 1:200. RyR: Sigma Aldrich, 1:1000. Ca$_V\beta_2$: Alomone, 1:200. Rab7: Cell Signaling Technology, 1:100. Rab5: Cell Signaling Technology, 1:200. Lamp1: Developmental Studies Hybridoma Bank, created by the NICHD of the NIH and maintained at The University of Iowa, Department of Biology, Iowa City, IA 52242, 1:100. Secondary antibodies (Thermofisher) were used at a dilution of 1:1000.

## Confocal microscopy

Cells were plated onto 35 mm MatTek imaging dishes (MatTek Corporation). Images were captured on a Nikon A1RMP confocal microscope with a 40x oil immersion objective (1.3 N.A.). CFP, Alexa-488, YFP, and Alexa-647 were imaged using 458, 488, 514 and 639 nm laser lines, respectively.

## Pulldown assays

Transfected HEK293 cells cultured in 60 mm dishes were harvested in PBS, centrifuged at 2,000 g (4°C) for 5 min, and the pellet resuspended in RIPA lysis buffer containing (mM): 150 NaCl, 20 Tris HCl, 1 EDTA, 0.1% (wt vol$^{-1}$) SDS, 1% Triton X-100, 1% sodium deoxycholate, and supplemented with protease inhibitor mixture (10 μL mL$^{-1}$, Sigma Aldrich), 1 PMSF, 2 N-ethylmaleimide,. 05 PR-619 deubiquitinase inhibitor (LifeSensors). Cells were lysed on ice for 1 hr with intermittent vortexing and centrifuged at 10,000 g for 10 min (4°C). The soluble lysate collected and protein concentration determined with the bis-cinchonic acid protein estimation kit (Pierce Technologies).

For $Ca_V\beta_{1b}$ pulldowns, lysates were precleared with 10 µL of protein A/G sepharose beads (Rockland) for 1 hr at 4°C and then incubated with 2 µg anti-$Ca_V\beta_1$ antibody (UC Davis/NIH NeuroMab Facility, clone N7/18) for 1 hr at 4°C. Equivalent amounts of protein were then added to spin columns with 25 µL equilibrated protein A/G sepharose beads and rotated overight at 4°C. Immunoprecipitates were washed a total of five times with RIPA buffer and then eluted with 30 µL elution buffer (50 mM Tris, 10% (vol vol$^{-1}$) glycerol, 2% SDS, 100 mM DTT, and 0.2 mg mL$^{-1}$ bromophenol blue) at 55°C for 15 min. For $\alpha_{1C}$ pulldowns, lysates were added to spin columns containing 10 µL of equilibrated RFP-trap agarose beads, rotated at 4°C for 1 hr, and then washed/eluted as described above. Proteins were resolved on a 4–12% Bis Tris gradient precast gel (Life Technologies) in MOPS-SDS running buffer (Life Technologies) at 200 V constant for ~1 hr. Protein bands were transferred by tank transfer onto a polyvinylidene difluoride (PVDF, EMD Millipore) membrane in transfer buffer (25 mM Tris pH 8.3, 192 mM glycine, 15% (vol/vol) methanol, and 0.1% SDS). The membranes were blocked with a solution of 5% nonfat milk (BioRad) in Tris-buffered saline-tween (TBS-T) (25 mM Tris pH 7.4, 150 mM NaCl, and 0.1% Tween-20) for 1 hr at RT and then incubated overnight at 4°C with primary antibodies ($Ca_V\beta_1$, UC Davis/NIH NeuroMab Facility. Actin, Sigma Aldrich) in blocking solution. The blots were washed with TBS-T three times for 10 min each and then incubated with secondary horseradish peroxidase-conjugated antibody for 1 hr at RT. After washing in TBS-T, the blots were developed with a chemiluminiscent detection kit (Pierce Technologies) and then visualized on a gel imager. Membranes were then stripped with harsh stripping buffer (2% SDS, 62 mM Tris pH 6.8, 0.8% ß-mercaptoethanol) at 50°C for 30 min, rinsed under running water for 2 min, and washed with TBST (3x, 10 min). Membranes were pre-treated with 0.5% glutaraldehyde and re-blotted with anti-ubiquitin (VU1, LifeSensors) as per the manufacturers' instructions.

### Calcium imaging

DRG neurons were washed twice in basal solution containing (mM): 145 NaCl, 5 KCl, 2 $CaCl_2$, 1 $MgCl_2$, one sodium citrate, 10 HEPES, 10 D-glucose, pH 7.4, and incubated in the same solution containing 5 uM fura-2 with 0.05% Pluronic F-127 detergent (Life Technologies) for 1 hr at 37°C, 5% $CO_2$. Afterwards, cells were washed twice in same solution and placed on an inverted Nikon Ti-eclipse microscope with a Nikon Plan fluor 20x objective (0.45 N.A.). Fura-2 measurements were recorded at excitation wavelengths of 340 and 380 nm using EasyRatioPro (HORIBA Scientific). DRG neurons were depolarized with a solution in which NaCl was reduced to 110 mM and KCl increased to 40 mM.

Pancreatic β-cells were imaged with a similar protocol. Cells were maintained in a basal KRBH solution composed of (mM): 134 NaCl, 3.5 KCl, 1.2 $KH_2PO_4$, 0.5 $MgSO_4$, 1.5 $CaCl_2$, 5 NaHCO3, 10 HEPES, 2.8 D-glucose, pH 7.4. Stimulation solutions included either 16.8 mM glucose or 40 mM KCl, with NaCl concentrations adjusted accordingly to balance osmolarity with KRBH solution.

### Data and statistical analysis

Data were analyzed off-line using FloJo, PulseFit (HEKA), Microsoft Excel, Origin and GraphPad Prism software. Statistical analyses were performed in Origin or GraphPad Prism using built-in functions. Statistically significant differences between means ($p < 0.05$) were determined using Student's *t* test for comparisons between two groups or one-way ANOVA for three groups, with Tukey's post-hoc analysis. Data are presented as means ± s.e.m.

## Acknowledgements

We thank Dr. Papiya Choudhury and Ming Chen for excellent technical support; the laboratory of Dr. Domenico Accili for assistance with pancreatic β-cells; Drs. Theanne Griffith and Chi-Kun Tong from the laboratory of Dr. Ellen Lumpkin for assistance with dorsal root ganglion neurons; Drs. K Satome, A Sobolevsky and E Cao for the gift of BacMam vectors. The work was supported by NIH grants R01 HL142111 and R01 GM107585 (to HMC) and F31 DK118866 (to TJM). Flow cytometry experiments were performed in CCTI Flow Cytometry Core, supported in part by the NIH (S10RR027050). Confocal images were collected in the HICC Confocal and Specialized Microscopy Shared Resource, supported by the NIH (P30 CA013696). Isothermal Titration Calorimetry measurements were performed at the Columbia University Precision Biomolecular Characterization Facility.

## Additional information

### Funding

| Funder | Grant reference number | Author |
| --- | --- | --- |
| National Institutes of Health | R01 HL142111 | Henry M Colecraft |
| National Institutes of Health | R01 GM107585 | Henry M Colecraft |
| National Institutes of Health | F31 DK118866 | Travis Morgenstern |
| National Institutes of Health | S10RR027050 | Travis Morgenstern<br>Jinseo Park<br>Qing R Fan<br>Henry M Colecraft |
| National Institutes of Health | P30 CA013696 | Travis Morgenstern<br>Jinseo Park<br>Qing R Fan<br>Henry M Colecraft |

The funders had no role in study design, data collection and interpretation, or the decision to submit the work for publication.

### Author contributions

Travis J Morgenstern, Conceptualization, Data curation, Formal analysis, Supervision, Funding acquisition, Investigation, Writing—original draft, Writing—review and editing; Jinseo Park, Conceptualization, Data curation, Formal analysis, Investigation, Methodology, Writing—original draft, Writing—review and editing; Qing R Fan, Resources, Methodology, Writing—review and editing; Henry M Colecraft, Conceptualization, Supervision, Funding acquisition, Investigation, Methodology, Writing—original draft, Writing—review and editing

### Author ORCIDs

Travis J Morgenstern (ID) https://orcid.org/0000-0003-2634-8470
Qing R Fan (ID) http://orcid.org/0000-0002-9330-0963
Henry M Colecraft (ID) https://orcid.org/0000-0002-2340-8899

### Ethics

Animal experimentation: This study was performed in strict accordance with the recommendations in the Guide for the Care and Use of Laboratory Animals of the National Institutes of Health. All of the animals were handled according to approved institutional animal care and use committee (IACUC) protocols (#A3007-01) of Columbia University. Primary cultures of adult guinea pig heart ventricular cells were prepared in accordance with the guidelines of Columbia University Animal Care and Use Committee protocols (AC-AAAS5410). All surgery was performed under isoflurane anesthesia, and every effort was made to minimize suffering.

### Decision letter and Author response

Decision letter https://doi.org/10.7554/eLife.49253.016
Author response https://doi.org/10.7554/eLife.49253.017

## Additional files

### Supplementary files

• Transparent reporting form
DOI: https://doi.org/10.7554/eLife.49253.014

### Data availability

All data generated or analysed during this study are included in the manuscript and supporting files.

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
