## [Decision Letter]

Thank you for submitting your article "A potent voltage-gated calcium channel inhibitor engineered from a nanobody targeted to auxiliary Ca_V_β subunits" for consideration by *eLife*. Your article has been reviewed by three peer reviewers, including Mark T Nelson as the Reviewing Editor and Reviewer #1, and the evaluation has been overseen by Olga Boudker as the Senior Editor. The following individual involved in review of your submission has agreed to reveal their identity: Kurt Beam (Reviewer #2).

The reviewers have discussed the reviews with one another and the Reviewing Editor has drafted this decision to help you prepare a revised submission.

Summary:

In this manuscript, Morgenstern et al. describe the development of a genetically encoded, nanobody-based system that inhibits the activity of high-voltage-activated calcium channels (Ca_V_1/Ca_V_2) without affecting the activity of low-voltage-activated calcium channels. To accomplish this, they fused a nanobody isolated from an antibody raised against Ca_V_β subunits (β1-4)-which was functionally inert alone-to the catalytic domain of the E3 ubiquitin ligase, Nedd4L. They then showed that this fusion protein, termed Ca_V_-aβlator, was capable of completing eliminating Ca_V_1/Ca_V_2 currents of reconstituted channel complexes in HEK cells and endogenous channels in cardiomyocytes, dorsal root ganglion neurons and pancreatic β-cells. They further demonstrated that Ca_V_-aβlator produced its inhibitory effects on Ca_V_1.2 channels by promoting their ubiquitination-dependent accumulation in Rab-7-positive late endosomes without promoting ubiquitin-dependent degradation of channel subunits. Further development of this system could lead to selective targeting of specific cell types or channels with different subunit composition. Overall, this elegantly conceived study makes excellent use of appropriate experimental approaches to convincingly demonstrate the operation and potential adaptability of their system.

Essential revisions:

1) It is difficult to make sense of results presented in Figure 3B and C without more stage-setting. Specifically, the absence of an effect of nb.F3-Nedd4L on Ca_V_β subunit expression runs counter to the expectation that Nedd4L, given its function as a ubiquitin ligase (and the fact that it binds directly to Ca_V_β), might cause Ca_V_β degradation. This becomes clear in subsequent Results sections, but it would be helpful to provide more context by setting up these earlier experiments as a test of whether Nedd4L acts broadly through degradation or via some other mechanism. The absence of an effect of nb.F3-Nedd4L on Ca_V_β subunit expression would then support the sub-conclusion that nb.F3-Nedd4L acts through a mechanism independent of ubiquitin-dependent degradation. It may be a minor point, but there's no good reason to be coy about the mechanism if doing so might leave some readers scratching their heads.

2) A cartoon showing the overall mechanism of action of Ca_V_-aβlation would be helpful.

3) Interestingly, in addition to localizing to the plasma membrane, β1 and β3 subunits, but not β2 or β4 subunits, also showed substantial accumulation in nuclear membranes. Do the authors have any thoughts on what causes this differential distribution and how it might be exploited in the context of their system?

4) Specific concerns about Figure 5. The text states that the data in Figure 5 show that the infection of cardiomyocytes with F3-Nedd4L causes the ubiquitination of Ca_V_1.2 and Ca_V_ß (I agree) and causes the channel complex to re-locate away from dyads and to become associated with Rab7. The latter appears unconvincing: the co-localization with RyR2 in the control myocytes is not robust (Figure 5C) and the distribution pattern of Ca_V_1.2 in the F3-Nedd4L infected cells is very different in Figure 5C (a scattered pattern, which is supposed to show movement away from dyads) and in Figure 5D (concentrated near the sides of the myocyte, which is supposed to show association with Rab7). These imaging data should be removed.

5) The data support the notion that the nanobody interacts with an epitope conserved among Ca_V_βs, thus allowing for a single construct with pan-Ca_V_ β binding. This is a nice feature for the first pass of experiments but much of the downstream utility/promise for the approach would require isoform specific nanobody development. While this is obviously theoretically possible, its notable that despite using two Ca_v_ β isoforms for nanobody production, the best performing hit is nonselective. The authors could do better in pointing out this area of future optimization for the reader.

6) The results are presented in a way that mostly ignores complexities of likely importance. In particular, if this technique is to be used either therapeutically or as an investigative tool, one would want to know something about the time course of the effect and about whether it would be possible to regulate the extent of action of the nanobody complex. It would be a major undertaking to obtain such information. Nonetheless, it would useful to discuss these potential complications.

---

## [Author Response]

Essential revisions:1) It is difficult to make sense of results presented in Figure 3B and C without more stage-setting. Specifically, the absence of an effect of nb.F3-Nedd4L on Ca_V_β subunit expression runs counter to the expectation that Nedd4L, given its function as a ubiquitin ligase (and the fact that it binds directly to Ca_V_β), might cause Ca_V_β degradation. This becomes clear in subsequent Results sections, but it would be helpful to provide more context by setting up these earlier experiments as a test of whether Nedd4L acts broadly through degradation or via some other mechanism. The absence of an effect of nb.F3-Nedd4L on Ca_V_β subunit expression would then support the sub-conclusion that nb.F3-Nedd4L acts through a mechanism independent of ubiquitin-dependent degradation. It may be a minor point, but there's no good reason to be coy about the mechanism if doing so might leave some readers scratching their heads.

We thank the reviewer for the constructive feedback. In the original manuscript, we prefaced the results presented in Figure 3 with a reference to our previous work which demonstrated that targeting the catalytic domain of the E3 ligase CHIP to YFP-tagged potassium or calcium channels using a GFP-nanobody selectively reduced their surface density without promoting degradation. In response to the reviewer’s concern we have changed the order of presentation in Figure 3 to better set the stage to communicate the results. Principally, we now present our examination of the impact of Ca_V_-aβlator on Ca_V_β expression first (original Figure 3A-C is now swapped with Figure 3D-F in the revised manuscript) and have added the following sentence:

“Given the classical role of E3 ubiquitin ligases in mediating degradation of target proteins, we first assessed if nb.F3-Nedd4L affected total Ca_V_β expression (Figure 3A, B).”

2) A cartoon showing the overall mechanism of action of Ca_V_-aβlation would be helpful.

We agree and have included such a cartoon in Figure 5H.

3) Interestingly, in addition to localizing to the plasma membrane, β1 and β3 subunits, but not β2 or β4 subunits, also showed substantial accumulation in nuclear membranes. Do the authors have any thoughts on what causes this differential distribution and how it might be exploited in the context of their system?

We agree with the reviewer that differential targeting of Ca_V_β isoforms to distinct sub-cellular compartments is interesting. However, we wish to emphasize that these are exemplar images obtained from overexpressing Ca_V_βs in a heterologous system. While the approach was sufficient for us to confirm binding of nanobodies to Ca_V_βs inside cells, we believe it is prudent to not over-interpret the potential basis or functional relevance of the differential Ca_V_β distribution from these data. While Ca_V_β subunits have been reported to have differential distributions and binding partners, that is not the focus of our current work.

4) Specific concerns about Figure 5. The text states that the data in Figure 5 show that the infection of cardiomyocytes with F3-Nedd4L causes the ubiquitination of Ca_V_1.2 and Ca_V_ß (I agree) and causes the channel complex to re-locate away from dyads and to become associated with Rab7. The latter appears unconvincing: the co-localization with RyR2 in the control myocytes is not robust (Figure 5C) and the distribution pattern of Ca_V_1.2 in the F3-Nedd4L infected cells is very different in Figure 5C (a scattered pattern, which is supposed to show movement away from dyads) and in Figure 5D (concentrated near the sides of the myocyte, which is supposed to show association with Rab7). These imaging data should be removed.

We thank the reviewer for this comment and have worked to make the data presented in Figure 5 clearer and more compelling. We have included magnifications of each image using a high-zoom box, that more clearly conveys the differential co-localization of α_1C_/RyR2 and α_1C_/Rab7 under the different conditions. This change has been described in the legend to Figure 5. Together with the quantitative Pearson’s correlation coefficient analyses, we believe these images affirm our finding that Ca_V_-aβlator predominantly directs Ca_V_1.2 to Rab7-positive late endosomes.

5) The data support the notion that the nanobody interacts with an epitope conserved among Ca_V_βs, thus allowing for a single construct with pan-Ca_V_ β binding. This is a nice feature for the first pass of experiments but much of the downstream utility/promise for the approach would require isoform specific nanobody development. While this is obviously theoretically possible, its notable that despite using two Ca_v_β isoforms for nanobody production, the best performing hit is nonselective. The authors could do better in pointing out this area of future optimization for the reader.

We agree with the reviewer that development of isoform-specific nanobodies would be important to realize the full potential of this approach. From our llama immunization we have obtained a number of nanobodies and are systematically working our way through to find potential isoform-specific variants. This is ongoing work that is beyond the scope of the current manuscript. We have added the following sentence to address this important future direction:

“However, this capability may be readily achieved with Ca_V_-aβlators directed towards particular Ca_V_β isoforms. A challenge to realize this possibility is the development of Ca_V_β isoform-specific nanobodies which should be feasible given that there is sequence divergence among Ca_V_βs outside the conserved src homology 3 (SH3) and guanylate kinase (GK) domains.”

6) The results are presented in a way that mostly ignores complexities of likely importance. In particular, if this technique is to be used either therapeutically or as an investigative tool, one would want to know something about the time course of the effect and about whether it would be possible to regulate the extent of action of the nanobody complex. It would be a major undertaking to obtain such information. Nonetheless, it would useful to discuss these potential complications.

We concur. We have integrated these points into our Discussion:

“For this purpose, it may be desirable to generate Ca_V_-aβlator versions whose time course and extent of action could be tuned by either a small molecule or light. Indeed, this is a focus of our ongoing work.”